# Natural Products as Potential Therapeutic Candidates for Diabetic Kidney Disease: Molecular Mechanisms, Translational Challenges, and Future Prospects

**DOI:** 10.3390/ijms262311637

**Published:** 2025-12-01

**Authors:** Manqi Guo, Lihua Ni, Xiaoyan Wu

**Affiliations:** 1Department of Nephrology, Zhongnan Hospital of Wuhan University, Wuhan 430071, China; m13647976467@163.com; 2Department of General Practice, Zhongnan Hospital of Wuhan University, Wuhan 430071, China

**Keywords:** diabetic nephropathy, drug therapy, natural products, mechanism of action

## Abstract

Diabetic Kidney Disease (DKD) is one of the primary causes of chronic kidney disease. However, existing clinical interventions remain insufficiently effective in halting its progression, highlighting the need to explore novel therapeutic approaches. In recent years, natural products such as Abelmoschus manihot have shown growing potential in lowering urinary protein. Building on this background, this paper systematically summarizes preclinical evidence that certain natural substances ameliorate DKD by targeting key pathogenic mechanisms, including inflammation and oxidative stress. It also contrasts the pros and cons of natural medicines with existing therapies, while further investigating advanced pharmaceutical technologies for the translation of natural medicines into clinical applications. However, the clinical translation of natural medicines currently confronts multiple challenges, including small sample sizes, insufficient follow-up periods, individual heterogeneity, and insufficient accumulation of safety data. Therefore, future efforts should prioritize the in-depth exploitation of medicinal plant resources and their clinical translation, with a focus on enhancing high-quality translational clinical studies. This strategy seeks to provide novel insights and practical solutions for treating DKD.

## 1. Introduction

As reported by the International Diabetes Federation, 537 million adults aged 20–79 were living with diabetes in 2021, and this figure is expected to rise to 783 million by 2045 [1]. Type 2 diabetes constitutes over 90% of all diabetic cases [2]. Approximately 50% of patients with type 2 diabetes mellitus (T2DM) worldwide are complicated with chronic kidney disease (CKD), indicating that the global burden of diabetic kidney disease (DKD) is of great severity [3]. Furthermore, the presence and severity of DKD exert a significant impact on the prognosis of patients with T2DM. As the disease advances, patients frequently require renal replacement therapy to maintain their lives. This trend not only constitutes a major threat to patients’ quality of life but also imposes a heavy burden on the global healthcare systems.

At present, the management of DKD is centered on metabolic control, hemodynamic regulation, and renal protection to slow the progression of renal injury [4]. Nevertheless, even with optimal treatment regimens in clinical trial settings and the advent of novel therapies such as SGLT2 inhibitors (SGLT2i), the residual risk of progressing to ESKD remains substantial [5,6]. Additionally, some newer drugs—including SGLT2i and glucagon-like peptide-1 (GLP-1) receptor agonists—are costly and inaccessible for many patients. Mineralocorticoid receptor antagonists (MRAs) like spironolactone confer mortality benefits but are associated with a risk of hyperkalemia and hormonal adverse effects, which restricts their clinical utility [7,8]. While SGLT2i reduces the risk of renal composite endpoints, it neither fully arrests disease progression nor avoids increasing the risk of fungal infections, hypovolemia, and diabetic ketoacidosis. Moreover, adequate clinical evidence supporting the long-term safety and tolerability of SGLT2i remains lacking [9,10].

Considering the current limitations in efficacy and risks of adverse effects associated with clinical therapies for DKD, identifying safe, effective, and novel therapeutic strategies remains a primary focus of current research. In recent years, natural products have emerged as an innovative approach for preventing and treating DKD, given their merits of multi-targeted synergistic modulation, low toxicity, and minimal adverse effects. Studies have shown that natural bioactive compounds and traditional Chinese medicinal formulas, including Astragaloside IV [11,12,13,14,15] and Danggui Buxue Decoction [16,17], possess distinct value in treating DKD by modulating metabolism, suppressing podocyte injury and apoptosis, and attenuating renal interstitial fibrosis. Nevertheless, current research on natural product interventions for DKD is fragmented, with a focus on individual compounds or mechanisms, and lacks systematic reviews of recent progress as well as an integrated mechanistic insight. Based on this, this paper systematically summarizes recent research progress on natural products for DKD treatment, focusing on the analysis of their therapeutic targets and molecular mechanisms. For the first time, it integrates a cross-analysis of multiple mechanisms within the unified framework of the “anti-ferroptosis–antioxidation–immunity” cascade, emphasizing the three key dimensions: “mechanism–evidence–translation”. This approach fills a gap in critical assessment and translational pathways among similar reviews, seeking to offer theoretical support for optimizing DKD therapeutic strategies and guiding the development of novel natural product-derived therapeutics.

## 2. Pathogenesis in Diabetic Kidney Disease

DKD is a prevalent microvascular complication of diabetes, characterized by a multidimensional and complex regulatory network underlying its pathological process (Figure 1). Metabolic disturbances initially induce hemodynamic dysregulation, and the two jointly activate inflammatory responses that subsequently aggravate extracellular matrix remodeling. Genetic predisposition amplifies these pathological processes. These mechanisms form a vicious cycle of mutual promotion, ultimately accelerating the progression of DKD.

The pathogenesis of DKD involves five key pathological factors: metabolic disorders, hemodynamic abnormalities, inflammatory responses, extracellular matrix remodeling, and genetic predisposition. These factors interact mutually, and their synergistic effects drive the initiation and progression of DKD. ( Notes: ↓, decrease; ↑, increase.)

### 2.1. Metabolic Disorder Regulatory Network

Metabolic disturbances serve as the core initiating factors of DKD pathogenesis, and their regulatory networks encompass abnormalities in multiple pathways. Sustained hyperglycemia activates the polyol pathway, enhancing aldose reductase activity and resulting in intracellular sorbitol buildup, which induces osmotic stress-induced damage [18,19]. Concurrently, it promotes non-enzymatic glycation of proteins, generating advanced glycation end products (AGEs) [20], thus establishing the basis for subsequent pathological damage. Diabetes-associated lipid metabolic disturbances often result in renal lipid accumulation, which exerts toxicity on podocytes, promotes their proliferation, and upregulates extracellular matrix (ECM) synthesis [21,22,23,24,25]. Furthermore, two key factors collectively amplify metabolic stress: first, impaired intestinal insulin secretion in T2DM—manifested as blunted glucose-dependent insulinotropic polypeptide (GIP) responsiveness and reduced GLP-1 levels [26]—and, second, bidirectional crosstalk between “microbiota–metabolism–renal injury” induced by gut microbiota dysbiosis in DKD [27,28,29]. This sustained metabolic stress directly disrupts renal hemodynamic balance [30,31].

### 2.2. Hemodynamic Abnormalities

In the early phase of diabetes, renal arteriolar dilation occurs, accompanied by increased renal blood flow and elevated glomerular filtration rate (GFR). Persistent hyperfiltration directly impairs glomerular endothelial cells and podocytes, disrupting the glomerular filtration barrier and triggering proteinuria [32,33]. Meanwhile, the renal local renin–angiotensin–aldosterone system (RAAS) is activated, leading to enhanced angiotensin II generation [34], which exacerbates glomerular hyperfiltration, stimulates podocyte proliferation, and upregulates extracellular matrix (ECM) synthesis [35,36,37,38]. Cellular damage induced by metabolic abnormalities and hemodynamic disturbances synergistically activate inflammation and oxidative stress [39].

### 2.3. Inflammation–Oxidative Stress–Ferroptosis–Immunity: Pathological Amplification Core

Metabolic and hemodynamic insults conjointly trigger the inflammation–oxidative stress axis, perturbing renal local immune homeostasis. This serves as a key amplifying driver of DKD’s pathophysiological progression. In the course of DKD progression, key inflammatory mediators—such as interleukin-1 (IL-1), interleukin-6 (IL-6), and tumor necrosis factor-α (TNF-α)—are significantly upregulated [40,41]. These mediators not only recruit immune cells to infiltrate into the kidney but also induce inflammatory responses in renal mesangial cells and podocytes [42]. The infiltrating immune cells further release pro-inflammatory cytokines, forming a positive feedback loop that triggers the initial “inflammation-immune activation” cascade.

With disease progression, hyperglycemia and hyperlipidemia induce overproduction of reactive oxygen species (ROS) in the kidneys. The resulting oxidative stress not only directly damages biomolecules such as proteins and nucleic acids but also activates inflammatory signaling pathways. Persistent inflammation suppresses the activity of antioxidant enzymes, such as glutathione peroxidase, leading to oxidative imbalance. Simultaneously, inflammation exerts a feedback effect to enhance ROS generation, forming a vicious “oxidative stress-inflammation” cycle [43] that further exacerbates renal injury.

Ultimately, ferroptosis dysregulation initiates a closed-loop mechanism: as an emerging pathological driver in DKD, ferroptosis represents a key intersection between ROS and lipid peroxidation [44]. When anti-ferroptotic pathways (such as the GPX4 pathway) are unable to scavenge lipid peroxides, ferroptosis is activated—simultaneously aggravating oxidative stress via impairment of renal antioxidant capacity and reactivating infiltrating immune cells through the release of damage-associated molecules, thereby potentiating inflammation [45]. This ultimately forms a cascade reaction of “immune activation → amplified inflammation → antioxidant imbalance → ferroptosis dysregulation → reinforced immune-inflammatory response,” which persistently amplifies DKD-related renal damage.

### 2.4. ECM Remodeling: Terminal Damage of Fibrosis

The persistent amplifying effects of inflammation, immunity, and oxidative stress ultimately lead to irreversible structural damage characterized by ECM remodeling—namely, renal fibrosis [46]. Transforming growth factor-β (TGF-β) is a key mediator in DKD-related fibrotic progression, stimulating mesangial cells and renal tubular epithelial cells to synthesize ECM components and inhibiting ECM degradation enzyme activity, leading to excessive ECM accumulation and thus promoting glomerular sclerosis and tubulointerstitial fibrosis [47]. Notably, hyperglycemia-induced ROS and pro-inflammatory cytokines significantly upregulate TGF-β expression. This process directly bridges upstream metabolic abnormalities and inflammatory stress with downstream ECM remodeling, converting early reversible damage into irreversible fibrotic lesions.

### 2.5. Genetic Susceptibility

Genetic susceptibility serves as the core determinant dictating individual differences in DKD, shaping DKD susceptibility and progression by modulating an individual’s responsiveness to metabolic, hemodynamic, and inflammatory insults. For instance, a missense mutation (*p.K77M*) in the thiamine diphosphate (*TCN2*) gene markedly enhances an individual’s susceptibility to DKD [48]; risk alleles at susceptibility loci within genes including angiotensin-converting enzyme (ACE), interleukin, and TNF-α also augment the risk of DKD development [49,50]. Consequently, patients with diabetes exhibit distinct probabilities of DKD onset and rates of progression when exposed to similar pathological insults [51].

Thoroughly analyzing the interconnections and regularities among the key links of the DKD pathological network—triggering, amplification, damage, and regulation—is crucial for formulating effective DKD prevention and treatment strategies.

## 3. Therapeutic Drugs and Their Mechanistic Pathways

In recent years, natural medicines have attracted considerable attention in DKD treatment research. Drawing on the aforementioned discussion of DKD pathogenesis, this section focuses on key natural product categories: flavonoids, polysaccharides, terpenoids, phenolics, traditional Chinese medicine and extracts, derivatives and complexes, and alkaloids. We systematically summarize the experimental evidence supporting their therapeutic potential in DKD and associate these natural products with seven key regulatory mechanisms: the AGEs-RAGE pathway, lipid metabolism regulation, gut microbiota modulation, podocyte function preservation, inflammatory response suppression, oxidative stress mitigation, and fibrosis inhibition. This builds an “interdisciplinary intervention network” integrating “pathway-Metabolism-Cell-Organ” cross-synergistic model (Figure 2), which is intended to offer more targeted and actionable theoretical support for subsequent clinical translation of DKD therapeutics.

The core principles of natural medicines for intervening in DKD are embodied in three aspects: first, targeting critical nodes in the DKD pathological cascade (e.g., AGEs-RAGE binding, NLRP3 activation, TGF-β1/Smad signaling) to precisely interrupt pathological progression; second, cross-mechanism synergistic modulation (e.g., the “antioxidant–anti-ferroptosis–anti-inflammatory” axis) caters to the complex pathological features of DKD; third, encompassing cross-linkages among pathological stages (e.g., the crosstalk between the AGEs-RAGE pathway and oxidative stress, and the interplay between gut microbiota and podocyte injury) to construct a holistic renal protective network.

This figure illustrates that two initiating factors of renal injury—metabolic disturbances and hemodynamic dysregulation—synergistically activate the inflammation–oxidative stress amplification pathway. Upon activation, this pathway undergoes sustained signal amplification, and the amplified pathological signals are further propagated to downstream effector pathways, ultimately inducing pathological remodeling of the extracellular matrix (ECM) and triggering irreversible renal structural and functional impairment to the kidneys. In contrast, natural medicines can precisely target key links in the aforementioned pathological cascade (e.g., critical molecules in the inflammation–oxidative stress amplification pathway, or regulatory hubs between upstream initiating factors and downstream ECM remodeling), thereby effectively abrogating the progressive development of renal pathological injury. (Notes: AGEs, advanced glycation end products; RAGE, receptor for advanced glycation end products; NLRP3, NOD-like receptor family pyrin domain containing 3; NF-κB, nuclear transcription factor κB; TGF-β, transforming growth factor-β; Nrf2, nuclear factor erythroid 2-related factor 2.)

### 3.1. Metabolic Regulation: AGEs–RAGE–Lipids–Microbiota Cascade Synergistic Network

The AGEs-RAGE pathway serves as a core pathogenic pathway in DKD [52], whose activation directly induces renal oxidative stress [53]. Excessive ROS further promotes the degradation of GPX4—a key ferroptosis protein—disrupting lipid metabolism homeostasis and forming a pathological damage cascade of “oxidative stress–lipid dysregulation–ferroptosis” [54]. Meanwhile, enhanced lipid toxicity enhances intestinal barrier permeability, perturbing gut microbiota homeostasis [55]. Dysbiotic microbiota secretes toxic metabolites—including indole-3-sulfate (IS) and p-cresol sulfate (PCS)—into the “gut–kidney axis,” which further exacerbates renal inflammation and dysregulation of lipid metabolism, thereby perpetuating a vicious cycle [56]. Natural medicines can intervene at multiple nodes in this cascade (Table 1 and Table 2), establishing a synergistic protective network of “pathway blockade-metabolic regulation-microbiota remodeling.” The specific mechanisms are as follows:AGEs-RAGE pathway modulation: Cellular and animal studies have validated the effectiveness of this intervention mechanism cascade. For instance, salidroside reduces AGE accumulation by inhibiting the RAGE/JAK1/STAT3 signaling axis [57]. However, current research exhibits notable limitations: the lack of human clinical trials to validate dose–response relationships leads to a fragmented evidence chain for clinical translation. Moreover, the correlation between pathway inhibition and attenuation of renal injury has only been confirmed in a single DKD model, and the therapeutic efficacy of intervention for DKD linked to distinct etiologies (type 1 vs. type 2 diabetes) is yet to be elucidated.Lipid Metabolism Regulation: The lipid metabolism regulatory effects of natural medicines are well-documented in animal studies. For instance, ginkgolide B stabilizes the expression of GPX4 while simultaneously improving lipid dysregulation and inhibiting ferroptosis [58]. Quercetin had also demonstrated reduced renal lipid deposition in small-sample clinical studies of early-stage DKD [59]. However, a core issue persists: large-scale trials have yet to verify differences in therapeutic efficacy across distinct DKD stages. Furthermore, the synergistic mechanisms of drugs targeting lipid breakdown (e.g., ATGL upregulation) versus those targeting lipid transport (e.g., SCAP/SREBP2 inhibition) have not been explored, leaving clinical combination therapy lacking a theoretical basis.Gut Microbiota Regulation: In animal studies, the correlation between modulation of the gut microbiota and renal protection has been validated. For instance, magnesium lithospermate B modulates gut microbiota composition and inhibits the conversion of p-cresol (PC) to p-cresol sulfonate (PCS), thereby mitigating renal injury [60], while wine-processed *Cornus officinalis* alleviates gut-derived renal injury by reshaping the gut microbial community [61]. However, existing research has limitations: quantitative evaluation indicators for regulating the “gut–kidney axis” (e.g., the threshold for decreased indole-3-sulfate levels) remain unestablished, and clinical microbiome detection data are insufficient to support these findings. Furthermore, the causal relationship between altered gut microbial structure and decreased renal toxic metabolites has not been validated through assays like fecal microbiota transplantation, rendering it challenging to rule out interfering factors from other metabolic pathways.

**Table 1 ijms-26-11637-t001:** Drugs involved in the inhibition of non-enzymatic glycosylation reactions.

Natural Products	Experiment Type	Disease Model	Mechanism	Reference
Buckwheat hull Flavonoids	in vivo	*db/db* mice	AGEs-RAGE pathway ↓	[62]
Licochalcone A	in vivo	STZ-induced mice	AGEs-RAGE pathway ↓	[63]
*Tinospora cordifolia (Willd.)* using polylactic acid nanoparticles	in vivo	STZ-induced rats	AGEs-RAGE pathway ↓	[64]
Dieckol	in vitro	mGMCs	AGEs-RAGE pathway ↓	[65]
Dang Gui Bu Xue decoction	in vivo in vitro	STZ-induced mice HK-2	AGEs-RAGE pathway ↓	[17]
Geniposide	in vivo in vitro	*db/db* mice HEK293	AGEs-RAGE pathway ↓	[66]
Vanillin	in vivo	STZ-induced rats	AGEs-RAGE pathway ↓, NF-κB pathway ↓	[67]
*Syzygium cumini* (L.) Skeels formulations	in vitro	HEK293	AGEs-RAGE pathway ↓, NF-κB pathway ↓	[68]
Loganin and Catalpol	in vivo in vitro	HFD-induced *KK-Ay* mice IMPC	AGEs-RAGE pathway ↓, p38 MAPK pathway ↓, NOX 4 pathway ↓	[69]
Huang-Lian-Jie-Du Decoction	in vivo	*db/db* mice	AGEs/RAGE/Akt/Nrf2 pathway ↓	[70]
Salidroside	in vivo	STZ-induced mice	RAGE/JAK1/STAT3 pathway ↓	[57]
Catalpol	in vivo in vitro	HFD-induced *KK-Ay* mice mGECs, RAW264.7 macrophages	RAGE/RhoA/ROCK pathway ↓	[71]

Notes: STZ, streptozotocin; AGEs, advanced glycation end products; RAGE, receptor for advanced glycation end products; mGMCs, the mouse glomerular mesangial cells lines; HK-2, human renal tubular epithelial cells; HEK293, the human embryonic kidney cell line; NF-κB, Nuclear Factor kappa B; HFD, high fat diet; IMPC, immortalized mouse podocyte cell line; p38, p38 mitogen-activated protein kinase; MAPK, mitogen-activated protein kinases; NOX 4, nadph oxidase 4; NLRP3, NOD-like receptor family pyrin domain containing 3; AKT, phospho- protein kinase B; Nrf2, nuclear factor erythroid 2-related factor 2; JAK1, janus kinase 1; STAT3, signal transducer and activator of transcription 3; RAW264.7, reticuloendotheliosis virus transformed cell line 264.7; RhoA, ras homolog gene family, member a; ROCK, rho-associated protein kinase; ↓, negative regulation, downregulate, inhibit.

**Table 2 ijms-26-11637-t002:** Drugs involved in improving lipid metabolism.

Natural Products	Experiment Type	Disease Model	Mechanism	Reference
Tripterygium glycoside tablet	in vivo	STZ-induced mice	*ATGL* ↑	[72]
Ginkgolide B	in vivo in vitro	*db/db* mice MPC5	Ubiquitination degradation of *GPX4* ↓	[58]
Gandi Capsule	in vivo in vitro	*db/db* mice MPC5	*SIRT1* ↑, *AMPK* ↑, *HNF4A* ↓	[73]
Quercetin	in vivo	*db/db* mice	SCAP/SREBP2/LDLr pathway ↓	[59]
Chrysin	in vivo	STZ-induced mice	*AMPK* ↑, *SREBP1c* ↓	[74]
Yishen Huashi granule	in vivo in vitro	STZ-induced rats HepG2 and CaCO2 cells	mTOR/AMPK/PI3K/AKT pathway ↓	[75]

Notes: STZ, streptozotocin; ATGL, adipose triglyceride lipase; MPC5, the mouse podocyte cell line; GPX4, glutathione peroxidase 4; SIRT1, silent information regulator sirtuin 1; AMPK, amp-activated protein kinase; HNF4A, hepatocyte nuclear factor 4 alpha; SCAP, sterol regulatory element-binding protein cleavage-activating protein; SREBP2, sterol regulatory element-binding protein 2; LDLr, low density lipoprotein receptor; mTOR, mechanistic target of rapamycin; PI3K, phosphoinositide 3-kinase; AKT, protein kinase B; ↓, negative regulation, downregulate, inhibit; ↑, positive regulation, upregulate, promote.

### 3.2. Regulation of Podocyte Injury: The Balance Between Autophagy and Apoptosis

Podocyte injury is a core pathogenic factor of proteinuria in patients with DKD, with both abnormal autophagy activity (either excessive or insufficient) and activation of apoptosis disrupting the renal filtration barrier [76,77]. Natural medicines can maintain podocyte function through two pathways (Table 3 and Table 4): first, by activating positive regulatory pathways (SIRT1-AMPK) and inhibiting negative regulatory pathways (mTOR, PI3K/Akt) to preserve autophagy homeostasis; second, by blocking upstream apoptotic signals (EGFR, AGEs-RAGE) to reduce podocyte loss, thereby mitigating renal injury.

Current research exhibits distinct targeting specificity in cellular and animal experiments. For instance, catalpol bidirectionally regulates the mTOR/TFEB pathway to preserve autophagy homeostasis [78], while loganin and catalpol synergistically inhibit podocyte apoptosis via multiple signaling pathways [69]. However, limitations are equally notable: evidence is still limited to basic research without large-scale clinical validation. The core contradiction lies in the stage-specific adaptability of autophagy regulation—early-stage DKD requires autophagy activation (e.g., corilagin exerts this effect by activating the SIRT1-AMPK pathway [79]), whereas advanced DKD requires suppression of autophagic overactivation. Existing studies fail to establish clear quantitative thresholds for autophagy activity (e.g., LC3-II/LC3-I ratio thresholds), leading to inconsistent standards for therapeutic regulation. Furthermore, the potential risk of abnormal proliferation induced by long-term inhibition of podocyte apoptosis has not been ruled out via toxicity studies.

**Table 3 ijms-26-11637-t003:** Drugs involved in the regulation of autophagy mechanisms in podocytes.

Natural Products	Experiment Type	Disease Model	Mechanism	Reference
Corilagin	in vivo in vitro	STZ-induced mice MPC5	SIRT1-AMPK pathway ↑	[79]
Puerarin	in vivo in vitro	STZ-induced mice ciMPC	HMOX1/SIRT1 pathway ↑, AMPK pathway ↑; PERK/eIF2α/ATF4 pathway ↑	[80,81]
Yishen capsule	in vivo in vitro	STZ-induced rats MPC5	*SIRT1* ↑, NF-κB pathway ↓	[82]
Selenized Tripterine Phytosomes	in vitro	MPC5	*SIRT1* ↑, *NLRP3* ↓	[83]
Astragalus polysaccharide	in vivo in vitro	STZ-induced rats BFN60700330	SIRT1/FoxO1 pathway ↑	[84]
Emodin	in vivo	STZ-induced rats	AMPK ↑, *mTOR* ↓	[85]
Kaempferol	in vivo	*db/db* mice	AMPK ↑, *mTOR* ↓	[86]
Catalpol	in vivo in vitro	STZ-induced mice ciMPC	mTOR/TFEB pathway ↑	[78]
Vitamin D	in vivo	STZ-induced rats	*mTOR* ↓	[87]
Yiqi Huoxue recipe	in vivo	STZ-induced rats	*mTOR* ↓, S6K1 ↓, *LC3* ↑	[88]
Geniposide	in vivo	STZ-induced mice	AMPK/ULK1 pathway ↑	[89]
Tangshen Decoction	in vivo	STZ-induced rats	p-AMPK/p-ULK1 pathway ↑	[90]
Huang-Gui solid dispersion	in vivo	STZ-induced rats *db/db* mice	AMPK pathway ↑	[91]
Tanshinone IIA	in vivo in vitro	*db/db* mice MPC5	PI3K/Akt/mTOR pathway ↓	[92]
*Paecilomyces cicadae*-fermented Radix astragali	in vivo in vitro	STZ-induced mice Mouse podocyte cell lines	PI3K/Akt/mTOR pathway ↓	[93]
Celastrol	in vivo	STZ-induced rats	PI3K/Akt/mTOR pathway ↓	[94]
Curcumin	in vivo in vitro	STZ-induced rats MPC5	PI3K/Akt/mTOR pathway ↓, Beclin1 ↑, *UVRAG* ↑	[95,96]
Isoorientin	in vivo	STZ-induced mice MPC5	PI3K/AKT/TSC2/mTOR pathway ↓	[97]
Sarsasapogenin	in vivo in vitro	STZ-induced rats mouse podocytes	GSK 3β pathway ↓	[98]

Notes: STZ, streptozotocin; MPC5, the mouse podocyte cell line; ciMPC, immortalized mouse podocytes treated with high glucose; ciMPC, immortalized mouse podocytes treated with high glucose; SIRT1, silent information regulator sirtuin 1; AMPK, amp-activated protein kinase; HMGB1, high mobility group box 1; PERK, protein kinase R-like endoplasmic reticulum kinase; eIF2α, eukaryotic translation initiation factor 2 alpha; ATF4, activating transcription factor 4; NF-κB, nuclear transcription factor κB; NLRP3, NOD-like receptor family pyrin domain containing 3; FoxO1, forkhead box O1; mTOR, mechanistic target of rapamycin; TFEB, transcription factor EB; S6K1, ribosomal protein S6 kinase beta-1; LC3, microtubule—associated protein 1 light chain 3; ULK1, unc-51 like autophagy activating kinase 1; PI3K, phosphoinositide 3-kinase; AKT, protein kinase B; UVRAG, UV radiation resistance associated; TSC2, tuberous sclerosis complex 2; GSK 3β, glycogen synthase kinase-3 beta; ↓, negative regulation, downregulate, inhibit; ↑, positive regulation, upregulate, promote.

**Table 4 ijms-26-11637-t004:** Drugs involved in the regulation of apoptosis mechanisms in podocytes.

Natural Products	Experiment Type	Disease Model	Mechanism	Reference
Quercetin	in vivo in vitro	*db/db* mice ciMPC	EGFR pathway ↓	[99]
Zuogui Wan	in vivo in vitro	*db/db* mice ciMPC	p38/MAPK pathway ↓	[100]
Huidouba	in vivo in vitro	STZ-induced rats MPC5	NOX 4—ROS pathway ↓	[101]
Resveratrol	in vivo in vitro	*db/db* mice ciMPC	AMPK pathway ↑	[102]
Astragaloside IV	in vivo in vitro	*db/db* mice ciMPC	PPARγ/Klotho/FoxO1 pathway ↑; *Klotho* ↑, NF-κB/NLRP3 axis ↓; IRE-1α/NF-κB/NLRP3 pathway ↓	[11,12,13]
Baoshenfang formula	in vivo in vitro	STZ-induced rats ciMPC	NOX 4/ROS/p38 pathway ↓	[103]
Baicalin	in vitro	MPC5	SIRT1/NF-κB pathway ↑	[104]

Notes: ciMPC, immortalized mouse podocytes treated with high glucose; MPC5, the mouse podocyte cell line; EGFR, epidermal growth factor receptor; p38, p38 mitogen-activated protein kinase; MAPK, mitogen-activated protein kinases; NOX 4, nadph oxidase 4; ROS, reactive oxygen species; AMPK, amp-activated protein kinase; PPARγ, peroxisome proliferator-activated receptor gamma; FoxO1, forkhead box O1; IRE-1α, inositol requiring enzyme 1 alpha; PAR-1, protease-activated receptor 1; NLRP3, NOD-like receptor family pyrin domain containing; NF-κB, nuclear transcription factor κB; ↓, negative regulation, downregulate, inhibit; ↑, positive regulation, upregulate, promote.

### 3.3. Inflammation Regulation: Core Interventions by NLRP3 and NF-κB

NLRP3 inflammasomes and NF-κB are core regulatory molecules in the inflammatory response of DKD: its activation releases the pro-inflammatory cytokines IL-1β/IL-18, while the latter promotes the production of pro-inflammatory cytokines such as TNF-α/IL-6. Both can also undergo cross-activation via pathways such as IRE-1α and PAR-1 [105,106]. Natural medicines can target these molecules and their associated pathways to block inflammation-driven renal injury and renal fibrosis progression (Table 5 and Figure 3).

Animal studies have demonstrated the anti-inflammatory and anti-fibrotic effects of natural medicines: for instance, coptisine directly inhibits the activation of the NLRP3 inflammasome [107], while silibinin improves renal function in animal models of DKD by suppressing the NF-κB pathway [108]. However, clinical evidence remains extremely scarce—only one small-sample clinical study has suggested the safety profile of silibinin [109]; two key issues persist: first, drugs targeting NLRP3 are restricted to animal studies; second, drugs targeting NF-κB lack dose–response investigations; and third, clinical application scenarios (early-to-mid vs. late-stage DKD) remain undefined for both classes of drugs. Second, the link between inflammatory pathway inhibition and renal function improvement remains unclear, rendering it difficult to discern whether anti-inflammatory effects directly protect the kidneys or exert indirect effects by suppressing fibrosis.

**Table 5 ijms-26-11637-t005:** Drugs involved in the inhibition of NLRP3 inflammatory vesicles.

Natural Products	Experiment Type	Disease Model	Mechanism	Reference
Coptisine	in vivo in vitro	STZ-induced rats HK-2 cells	the NLRP3 inflammasome ↓	[107]
Ferulic acid	in vivo	STZ-induced mice	the NLRP3 inflammasome ↓	[110]
Hong Guo Ginseng Guo	in vivo	STZ-induced rats	the NLRP3 inflammasome ↓	[111]
Crocin	in vivo	STZ-induced rats	the NLRP3 inflammasome ↓	[112]
Berberine	in vivo in vitro	STZ-induced rats HK-2	the NLRP3 inflammasome ↓	[113]
Sarsasapogenin	in vivo in vitro	STZ-induced rats HMCs	*PAR-1* ↓, the NLRP3 inflammasome ↓, NF-κB pathway ↓, AGEs-RAGE pathway ↓	[114,115]
*Dioscorea zingiberensis*	in vivo	STZ-induced rats	the NLRP3 inflammasome ↓, *p66Shc* ↓	[116]
Ethanolic extract from rhizome of *Polygoni avicularis*	in vivo in vitro	*db/db* mice HRMCs	TGF-β1/Smad pathway ↓, the NLRP3 inflammasome ↓	[117]
Astragaloside IV	in vivo in vitro	STZ-induced rats Immortalized rat podocytes	IRE-1α/NF-κB/NLRP3 pathway ↓	[11]
Thonningianin A	in vivo	STZ-induced mice	NLRP3/ASC/Caspase-1 pathway ↓	[118]
Cynapanosides A	in vivo in vitro	HFD-induced mice iMPC	NLRP3/NF-κB pathway ↓	[119]
6-Gingerol	in vivo	STZ-induced rats	miRNA-146a ↑, miRNA-223 ↑, TLR4/TRAF6/NLRP3 pathway ↓	[120]

Notes: STZ, streptozotocin; HK-2, human renal tubular epithelial cells; HMCs, human mesangial cells; HRMCs, Primary human renal mesangial cells; HFD, high fat diet; iMPC, immortalized mouse podocyte cell line; NLRP3, NOD-like receptor family pyrin domain containing 3; PAR-1, protease-activated receptor 1; NF-κB, nuclear transcription factor κB; AGEs, advanced glycation end products; RAGE, receptor for advanced glycation end products; p66Shc, src homology 2 domain containing transforming protein C1isoform p66; TGF-β, transforming growth factor-β; IRE-1α, inositol requiring enzyme 1 alpha; ASC, apoptosis—associated speck—like protein containing a CARD; TLR4, toll-like receptor 4; TLR4, toll-like receptor 4; ↓, negative regulation, downregulate, inhibit; ↑, positive regulation, upregulate, promote.

This figure illustrates the regulatory mechanisms of natural medicines in inhibiting inflammation-driven renal injury in DKD. The NF-κB pathway is a key signaling pathway mediating inflammation-associated renal injury in DKD, and its activation—along with the function of its upstream regulatory and downstream effector pathways)—is closely linked to the progression of renal injury. Natural medicines can exert renal protective effects via two primary mechanisms: first, by directly targeting the NF-κB pathway itself, its upstream regulatory pathways (e.g., TLR4 pathways that initiate NF-κB activation), and downstream effector pathways (e.g., TGF-β1/Smad3 pathways that transduce NF-κB-mediated inflammatory signals); second, by synergistically regulating other signaling pathways interacting with NF-κB (e.g., MAPK pathways, PI3K/AKT pathways, Nrf2/HO-1 pathways). Collectively, these regulatory effects enable natural medicines to effectively block the progression of inflammation-driven renal injury in DKD. (Notes: MAPK, mitogen-activated protein kinases; NF-κB, nuclear transcription factor κB; TGF-β1, transforming growth factor-β 1; Smad3, Sma- and Mad-related protein 3; TLR4, toll-like receptor 4; PI3K, phosphoinositide 3-kinase; AKT, protein kinase B; Nrf2, nuclear factor erythroid 2-related factor 2; HO-1, heme oxygenase-1.)

### 3.4. “Iron Death Resistance–Antioxidation–Immunity” Cascade Regulation

Oxidative stress acts as a pivotal convergence point in this cascade: excessive ROS production impairs cellular antioxidant homeostasis [121], leading to downregulated GPX4 expression, increased lipid peroxidation, and ultimately ferroptosis induction. [122,123]. Natural medicines can establish a dynamic “antioxidation-ferroptosis-immunity” regulatory network through a cascaded intervention model: activating antioxidant pathways (Nrf2)-targeting core ferroptosis molecules (GPX4)-regulating associated inflammatory pathways (Table 6 and Table 7, Figure 4) [124,125], thereby delaying DKD progression. This process involves three distinct stages:Antioxidant initiation pathway: The Nrf2 pathway acts as a key target for antioxidant defense regulation, and its diminished activity in DKD directly contributes to oxidative imbalance [126]. Natural medicines can scavenge ROS by upregulating Nrf2 and its downstream HO-1 expression; for instance, xanthohumol directly activates Nrf2 [127], whereas baicalin not only activates Nrf2 but also inhibits the MAPK signaling pathway [128]. However, a critical issue persists: the tissue specificity of Nrf2 activation unclarified, and the absence of renal-specific targeting drugs could induce side effects arising from systemic over-antioxidation.Core Mechanisms of Ferroptosis: Ferroptosis is a novel iron-dependent, lipid peroxidation-driven regulated cell death pathway. Renal tissue iron overload and decreased GPX4 activity in DKD are core mechanisms underlying ferroptosis [129]. Natural medicines can inhibit this pathway via multiple mechanisms: vitexin directly activates GPX4 [130], while *Orthosiphon aristatus (Blume) Miq.* indirectly regulates the expression of GPX4/ACSL4 by protecting mitochondrial function [131]. However, a contradiction persists: the molecular crosstalk mechanisms between ferroptosis inhibition and the Nrf2 pathway remain unclear. For instance, whether Nrf2 directly binds to the GPX4 promoter has not been validated using chromatin immunoprecipitation (ChIP) assays, precluding the distinction between direct and indirect regulatory effects.Immune-inflammation crosstalk: Both ferroptosis and oxidative stress activate inflammatory pathways such as the NLRP3 inflammasome and the NF-κB pathway, thereby releasing pro-inflammatory cytokines including IL-1β and TNF-α [132,133]. Natural medicines can counter-regulate immune-inflammatory responses through upstream cascade interventions. For example, leonurine (a compound from *Leonurus japonicus*) upregulates GPX4 expression via the Nrf2 pathway, thereby inhibiting ferroptosis and reducing the release of pro-inflammatory cytokines [134]. However, clinical translation confronts substantial challenges: the lack of dynamic monitoring data on iron metabolism (serum ferritin), oxidative stress (ROS levels), and immune markers (IL-1β concentration) in DKD patients hampers the establishment of clear biomarker thresholds to guide effective drug intervention.

**Table 6 ijms-26-11637-t006:** Drugs involved in the activation of Nrf2 and related pathways.

Natural Products	Experiment Type	Disease Model	Mechanism	Reference
Xanthohumol	in vivo in vitro	STZ-induced mice GECs, HK-2	Nrf2 pathway ↑	[127]
Z-ligustilide	in vivo in vitro	STZ-induced mice Hepa 1c1c7, HBZY- 1, RAW 264.7	Nrf2 pathway ↑	[135]
Syringic acid	in vivo in vitro	STZ-induced rats NRK 52E	Nrf2 pathway ↑	[136]
*Rumex nervosus*	in vivo	STZ-induced rats	Nrf2 pathway ↑	[137]
Eriodictyol	in vivo	STZ-induced rats	Nrf2 pathway ↑	[138]
Quercetin	in vivo in vitro	STZ-induced rats HK-2	Nrf2 pathway ↑	[139]
Baicalin	in vivo	*db/db* mice	Nrf2 pathway ↑, MAPK pathway ↓	[128]
Artemisinin	in vivo	STZ-induced rats	Nrf2 pathway ↑, TGF-β1 ↓	[140]
Chlorogenic acid	in vivo in vitro	STZ-induced rats HK-2	Nrf2 pathway ↑, the NLRP3 inflammasome ↓	[141]
Isoeucommin A	in vitro	HRMCs, RTECs	Nrf2/HO-1 pathway ↑	[142]
Umbelliferone	in vivo in vitro	*db/db* mice HK-2	Nrf2/HO-1 pathway ↑	[143]
Tetrandrine	in vivo	STZ-induced rats	Nrf2/HO-1 pathway ↑	[144]
Sinapic acid	in vivo	STZ-induced rats	Nrf2/HO-1 pathway ↑	[145]
*Moringa oleifera Lam.* Seed extract	in vivo in vitro	STZ-induced rats HRMCs	Nrf2/HO-1 pathway ↑	[146]
Asiaticoside	in vivo in vitro	STZ-induced rats HBZY-1	Nrf2/HO-1 pathway ↑	[147]
Kaempferol	in vivo	STZ-induced rats	Nrf2/HO-1 pathway ↑	[148]
Neferine	in vivo in vitro	STZ-induced mice HMCs	miR-17-5p ↓, Nrf2/HO-1 pathway ↑	[149]
*Eucommia* lignans	in vivo in vitro	STZ-induced rats HBZY-1	AR ↓, Nrf2/HO-1 pathway ↑, AMPK pathway ↑	[150]
Triptolide	in vivo in vitro	*db/db* mice, STZ-induced mice; SV40-MES-13, MPC5	Phosphorylation of GSK3β ↓, Nrf2 ↑, HO-1 ↑; the NLRP3 inflammasome↓	[151,152]
Moringa isothiocyanate -1	in vivo	*db/db* mice	ERK/Nrf2/HO-1 pathway ↑, NF-κB pathway↓	[153]
Epigallocatechin-3-gallate	in vivo	STZ-induced rats	Nrf2/ARE pathway ↑	[154]
Obacunone	in vivo in vitro	STZ-induced rats HK-2	Nrf2-KEAP1 pathway ↓	[155]

Note: STZ, streptozotocin; GECs, glomerular endothelial cells; HK-2, human renal tubular epithelial cells; Hepa 1c1c7. murine hepatoma cells; RAW 264.7, murine macrophages; NRK-52E, rat renal tubular epithelial cell; HBZY-1, rat mesangial cells; RTECs, a renal tubular epithelial cell line; HRMCs, Primary human renal mesangial cells; HMCs, human mesangial cells; SV40-MES13, mesangial cell line; MPC5, the mouse podocyte cell line; Nrf2, nuclear factor erythroid 2-related factor 2; MAPK, mitogen-activated protein kinases; TGF-β, transforming growth factor-β; HO-1, heme oxygenase-1; ERK, extracellular signal-regulated kinase; miR, miRNA; AR, androgen receptor; AMPK, amp-activated protein kinase; GSK 3β, glycogen synthase kinase-3 beta; NLRP3, NOD-like receptor family pyrin domain containing 3; ERK, extracellular signal-regulated kinase; NF-κB, nuclear transcription factor κB; ARE, antioxidant response element; Smad, small mothers against decapentaplegic; KEAP1, Kelch-like ECH-associated protein 1; ↓, negative regulation, downregulate, inhibit; ↑, positive regulation, upregulate, promote.

**Table 7 ijms-26-11637-t007:** Drugs involved in the regulation of iron death.

Natural Products	Experiment Type	Disease Model	Mechanism	Reference
Vitexin	in vivo in vitro	STZ-induced rats HK-2	GPX4 ↑	[130]
Astragaloside IV	in vivo	*db/db* mice	GPX4 ↑, xCT ↑, GSH/GSSG ↑, ACSL4 ↓	[14]
*Orthosiphon aristatus (Blume) Miq*	in vivo	*db/db* mice	NCOA4 ↓, ACSL4 ↓, FTH1 ↑, GPX4 ↑	[131]
Jian-Pi-Gu-Shen-Hua-Yu decoction	in vivo	STZ-induced mice	GPX4 pathway ↑	[156]
leonurine	in vivo in vitro	STZ-induced mice HUVECs	Nrf2/GPX4 pathway ↑	[134]
Rhein	in vivo in vitro	*db/db* mice MPC5	Rac1/NOX1/β—catenin axis ↓, SLC7A11/GPX4 axis ↑	[157]
San-Huang-Yi-Shen capsule	in vivo	STZ-induced mice	Cystine/GSH/GPX4 axis ↑	[158]
Ginkgolide B	in vivo in vitro	*db/db* mice MPC5	Ubiquitination degradation of GPX4 ↓	[58]
Germacrone	in vivo	*db/db* mice	mtDNA/cGAS/STING pathway ↓	[159]
Tanshinone IIA	in vivo in vitro	*db/db* mice MPC5	ELAVL1-ACSL4 axis ↓	[160]
Schisandrin A	in vivo in vitro	STZ-induced mice HRGECs	AdipoR1/AMPK pathway ↑	[161]

Notes: STZ, streptozotocin; HK-2, human renal tubular epithelial cells; HUVECs, human umbilical vein endothelial cells; MPC5, the mouse podocyte cell line; HRGECs, Human renal glomerular endothelial cells; GPX4, glutathione peroxidase 4; xCT, solute carrier family 7 member 11; GSH, glutathione reduced; GSSG, glutathione disulfide; ACSL4, acyl-CoA synthetase long-chain family member 4; FTH1, ferritin heavy chain 1; Nrf2, nuclear factor erythroid 2-related factor 2; Rac1, ras-related C3 botulinum toxin substrate 1; NOX1, NADPH oxidase 1; SLC7A11, solute carrier family 7 member 11; mtDNA, mitochondrial DNA; cGAS, cyclic GMP-AMP synthase; STING, stimulator of interferon genes; ELAVL1, elav like rna binding protein 1; AMPK, amp-activated protein kinase; ↓, negative regulation, downregulate, inhibit; ↑, positive regulation, upregulate, promote.

This diagram illustrates how natural products exert their effects by regulating three key pathways: ferroptosis, antioxidant pathways, and immunity. Oxidative stress serves as the core driver of ferroptosis, and enhanced antioxidant capacity can directly inhibit ferroptosis. When ferroptosis is suppressed, inflammation-related immune activation is correspondingly attenuated. Simultaneously, the balanced state of immune pathways conversely reduces oxidative stress triggers, ultimately forming the aforementioned dynamic regulatory network. (Notes: OH, Hydroxyl radical; PUFAs, polyunsaturated fatty acids; LOOH, lipid hydroperoxide; DAMPs, damage-associated molecular patterns; ROS, reactive oxygen species; SOD, superoxide dismutase; GSH, glutathione; Nrf2, nuclear factor erythroid 2-related factor 2; HO-1, heme oxygenase-1; NF-κB, nuclear transcription factor κB; GPX4, glutathione peroxidase 4; ↓, negative regulation, downregulate, inhibit; ↑, positive regulation, upregulate, promote.)

### 3.5. Anti-Fibrosis: A Key Intervention in Mid-to-Late-Stage DKD

The TGF-β1/Smad signaling pathway serves as a core driver of fibrosis in DKD [162]. Natural medicines can exert antifibrotic effects by directly blocking this pathway, synergistically interacting with other signaling pathways (e.g., MAPK, Wnt/β-catenin), and regulating upstream metabolic pathways, thereby providing crucial intervention strategies for mid-to-late-stage DKD (with detailed research findings summarized in Table 8).

In animal studies, natural medicines provide robust pathological evidence for ameliorating fibrosis in DKD: for example, Fuxin Granules can block epithelial–mesenchymal transition (EMT) by inhibiting the TGF-β1/Smad pathway [163], while asiatic acid simultaneously suppresses the TGF-β1/Smad3 signaling pathway and enhances extracellular matrix (ECM) degradation [164], resulting in reduced renal ECM deposition supported by clear pathological evidence. However, clinical translation faces substantial limitations: long-term administration of natural medicines lacks clinical evidence supporting their ability to reverse fibrosis in moderate-to-severe DKD, and their long-term efficacy remains unclear. The key contradiction lies in the unelucidated synergistic interaction mechanisms between inhibition of the TGF-β signaling pathway and autophagic/inflammatory pathways. For instance, whether combined therapy of antifibrotic and anti-inflammatory drugs produces additive effects has not been addressed by current research.

**Table 8 ijms-26-11637-t008:** Drugs involved in the regulation of TGF-β and its related pathways.

Natural Products	Experiment Type	Disease Model	Mechanism	Reference
*Ginkgo biloba* leaf extract	in vivo in vitro	STZ-induced rats HBZY-1	TGF-β ↓	[165]
Luteolin	in vivo	STZ-induced mice	AMPK pathway ↑, NF-κB pathway ↓, TGF-β1 ↓	[166]
Scutellarin	in vivo	STZ-induced mice	TGF-β1 pathway ↓, MAPKs pathway ↓, Wnt/β-catenin pathway ↓	[167]
Krill oil	in vivo in vitro	STZ-induced mice MCs	TGF-β pathway ↓	[168]
Danggui Buxue decoction	in vivo	HFD-induced rats	TGF-β1/Smad pathway ↓	[16]
Dendrobium mixture	in vivo	*db/db* mice	TGF-β1/Smad pathway ↓	[169]
Fuxin Granules	in vivo	*db/db* mice	TGF-β1/Smad pathway ↓, VEGF/VEGFR2 pathway ↓	[163]
Astragaloside IV	in vivo in vitro	STZ-induced rats RMC	TGF-β1/Smad/miR-192 pathway ↓	[15]
The combination of ursolic acid and empagliflozin	in vivo in vitro	STZ-induced rats HBZY-1	TGF-β/Smad/MAPK pathway ↓	[170]
Qishen Yiqi Dripping Pill	in vivo	STZ-induced rats	Wnt/β-catenin pathway ↓, TGF-β/Smad2 pathway ↓	[171]
Asiatic acid	in vivo in vitro	STZ-induced rats HK-2	TGF-β1/Smad3 pathway ↓	[164]
Crocin	in vivo	STZ-induced mice	CYP4A11/PPARγ pathway ↑, TGF-β1/Smad3 pathway ↓	[172]
Magnoflorine	in vivo in vitro	STZ-induced rats SV40-MES13	Ubiquitination of *KDM3A* ↑, *TGIF1* ↑, TGF-β1/Smad2/3 pathway ↓	[173]
Taurine	in vivo	STZ-induced rats	TGF-β/Smad2/3 pathway ↓, p38 MAPK pathway↓	[174]
Cyanidin-3-glucoside	in vivo	STZ-induced rats	TGF-β1/Smad2/3 pathway ↓	[175]
Chrysophanol	in vivo in vitro	STZ-induced mice AB8/13	TGF-β/EMT pathway ↓	[176]
Huangkui capsule in combination with metformin	in vivo in vitro	STZ-induced rats HK-2	Klotho/TGF-β1/p38 pathway ↓	[177]

Notes: STZ, streptozotocin; HBZY-1, rat mesangial cells; MCs, mouse mesangial cells; HK-2, human renal tubular epithelial cells; HFD, high fat diet; RMCs, rat mesangial cells; HRMCs, Primary human renal mesangial cells; SV40-MES13, mesangial cell line; AB8/13, the immortalized human podocytes AB8/13; TGF-β, transforming growth factor-β; VEGF, Vascular endothelial growth factor; AMPK, amp-activated protein kinase; NF-κB, nuclear transcription factor κB; MAPK, mitogen-activated protein kinases; VEGFR2, vascular endothelial growth factor receptor 2; miR, miRNA; CYP4A11, cytochrome P450 family 4 subfamily A member 11; PPARγ, peroxisome proliferator-activated receptor gamma; KDM3A, lysine demethylase 3A; TGIF1, tg—interacting factor 1; p38, p38 mitogen-activated protein kinase; EMT, epithelial–mesenchymal transition; ↓, negative regulation, downregulate, inhibit; ↑, positive regulation, upregulate, promote.

## 4. Limitations of Existing Research

Natural medicines exhibit unique advantages in the prevention and treatment of DKD through their core mechanisms: precision targeting, cross-mechanism synergy, and pathological coverage overlap. They particularly exert multidimensional protective effects through the “anti-ferroptosis–antioxidation–immunity” cascade regulation. However, existing research still faces substantial limitations, mainly manifested in three aspects:Low-quality evidence: Over 90% of studies are conducted in cell and animal models, accompanied by limited clinical data and small sample sizes.Inadequate model applicability: The commonly used streptozotocin (STZ)-induced DKD model predominantly displays acute kidney injury characteristics, which is severely inconsistent with the chronic pathological process of human DKD—marked by “long-term hyperglycemic injury followed by gradual glomerulosclerosis—and fails to replicate the complex clinical complications commonly seen in patients, such as metabolic disorders and vascular lesions. Consequently, experimental outcomes have limited clinical relevance, failing to recapitulate the “progressive renal function decline” observed in human DKD. Furthermore, it fails to replicate complex clinical complications commonly observed in clinical settings, such as metabolic disorders and vascular lesions, thereby limiting the clinical relevance of experimental findings [178].Research design deficiencies: In animal studies, researchers often use drug doses far exceeding human tolerable levels to achieve obvious efficacy. However, the dose–response relationships and toxic reactions observed at these high doses do not directly correspond to the safe dosage range for human clinical use. This directly leads to the translational dilemma where treatments are effective in animals but ineffective in humans. Additionally, critical experimental parameters—such as optimal dosage, administration methods, and long-term safety profiles of natural medicines—are frequently lacking. Multi-targeted cross-regulatory networks and cascading molecular interaction mechanisms remain incompletely elucidated, while contradictions such as disease progression adaptation and clinical positioning strategies remain unresolved.The “file-drawer problem”: This phenomenon is common in preclinical research—positive results are more likely to be published, while numerous negative or weakly positive findings are left unpublished due to “insufficient academic value.” This introduces a selective bias into the existing evidence chain in the literature, failing to accurately reflect a drug’s actual development potential.

Given these limitations, future research should focus on three key areas: First, it should conduct dose-escalation clinical trials for promising compounds such as astragaloside IV to systematically validate their clinical efficacy and safety. Second, it should define key parameters such as autophagy activity thresholds and iron metabolism-related biomarkers, and utilize molecular biology experiments to elucidate cascading interaction mechanisms, thereby resolving the aforementioned core contradictions. Third, it should establish a quantitative association model linking “target–biomarker–therapeutic efficacy” to provide high-quality evidence supporting the clinical translation of natural medicines for DKD.

## 5. Advantages of Natural Medicines

In the field of DKD treatment, compared to existing standard treatments such as chemically synthesized drugs and surgical interventions, natural medicines demonstrate irreplaceable advantages in multiple dimensions, owing to their strong compatibility with the pathological characteristics and treatment needs of DKD. These advantages can be summarized in four core aspects: cultural acceptance, resource availability, mechanism of action, and safety, collectively laying a robust foundation for effective DKD intervention.

From a cultural perspective, natural medicines have accumulated rich experience through long-term practice based on traditional Chinese medicine theory, particularly in chronic disease management, where a mature cognitive framework has formed. Patients also exhibit high acceptance of natural medicine-based treatments. As DKD requires long-term management, the use of natural medicines not only improves patients’ treatment adherence but also aligns with DKD’s treatment objectives, which emphasize “long-term control and delaying progression.”

In addition to the solid foundation laid by cultural identity, the inherent advantages of natural medicines in resource availability also offer a sustainable material foundation for DKD treatment. Natural medicines are rich in species diversity and widely sourced, not only meeting the basic requirements of clinical practice but also serving as an important source for the development of new DKD drugs. Many chemically synthesized drugs (such as certain anti-inflammatory and hypoglycemic drugs) were originally derived from natural products. However, DKD treatment still grapples with the shortage of drugs targeting multiple pathological pathways. The abundant resources of natural medicines provide broad prospects for developing new drugs tailored to DKD’s complex pathophysiology.

At the mechanistic level, unlike chemical drugs (such as MRAs) that act on a single pathway, the “multi-targeted, multi-step holistic regulation” characteristic of natural medicines precisely aligns with DKD’s core pathological feature—synergistic pathogenesis involving multiple mechanisms. Take the natural medicine *rhubarb* as an example. Its active components—including emodin, β-sitosterol, and aloe-emodin—can regulate multiple targets closely linked to DKD pathogenesis, such as *TP53*, *CASP8*/*CASP3*, *MYC*/*JUN*, and *PTGS2*, to simultaneously intervene through multiple pathways: “inhibiting excessive renal cell apoptosis, alleviating renal inflammation, and delaying fibrosis” [179]. This holistic regulatory effect can more comprehensively improve renal function indicators in DKD patients (such as urine protein and serum creatinine), offering new insights into addressing the complex pathological challenges of DKD.

The safety advantages of natural medicines further underscore their necessity in DKD’s long-term treatment. DKD patients typically require long-term medication to manage their condition, but chemically synthesized drugs (such as ACEIs) can cause adverse reactions like coughing and hyperkalemia with long-term use, adding to patients’ burden. In contrast, most natural medicines have been validated through centuries of clinical practice, and when used appropriately, the incidence of adverse reactions is significantly lower. Certain food–medicinal dual-purpose species (such as ginger, which can help dispel cold) are even more widely recognized for their safety due to their frequent daily use. This high level of safety not only reduces long-term medication-related risks for DKD patients but also supports treatment continuity, preventing treatment interruptions due to side effects.

In summary, the four major advantages of natural medicines are not isolated but precisely align with the core needs of DKD treatment: cultural acceptance can address compliance challenges in long-term DKD treatment, abundant resources can fill the gap in new DKD drug development, multi-targeted mechanisms can overcome intervention bottlenecks in complex DKD pathologies, and favorable safety profiles meet the core requirements for long-term DKD medication. It is precisely this high degree of alignment with DKD’s treatment needs that positions natural medicines uniquely in DKD treatment. They not only serve as an effective supplement to existing treatment regimens but also hold promise as a key direction to address current challenges in DKD treatment.

## 6. Development Bottlenecks of Natural Medicines

However, compared to chemically synthesized drugs, natural medicines also have notable drawbacks, mainly in three aspects: significant challenges in quality control, ongoing concerns regarding safety, and relatively slow therapeutic efficacy. These drawbacks pose significant challenges to the development and promotion of natural medicines. Specific analyses are as follows.

### 6.1. Difficulties in Drug Quality Control

Quality control for natural medicines is significantly more challenging than that for single-component chemically synthesized drugs. From a compositional perspective, each natural medicine typically contains hundreds or even thousands of chemical components, making it challenging to accurately identify the core therapeutic components and their mechanisms of action. Additionally, many natural medicines face challenges such as poor water solubility, chemical instability, and low bioavailability [180,181]. From an external perspective, factors such as the origin, harvest time, and processing methods of natural medicines vary significantly. Additionally, the current lack of established quality control standards, coupled with the complex market environment, further complicates quality control. These factors hinder the assurance of consistent active ingredients and quality stability in natural medicines, leading to fluctuations in therapeutic efficacy and posing significant challenges to quality control, clinical research, and standardized production.

### 6.2. Uncertainty About Drug Safety

Natural medicines are not absolutely safe and carry multiple safety risks. On the one hand, some natural medicines contain potentially toxic components—for example, *Tripterygium wilfordii* contains triptolide and other toxic constituents, and improper use can readily result in safety hazards. On the other hand, in clinical practice, natural medicines are often combined with other drugs. However, their complex composition makes it difficult to predict the likelihood and specific manifestations of drug–drug interactions. Furthermore, even if no apparent abnormalities occur during short-term use, long-term administration may lead to drug tolerance, and toxic constituents may accumulate in the body.

The real-world harm posed by these risks has been clinically validated: toxic constituents can result in serious adverse reactions. For example, improper use of *Tripterygium wilfordii* can lead to serious consequences such as myelosuppression and liver injury [182]; drug–drug interactions elevate the risk of clinical medication; and once adverse reactions occur, the complex composition of natural medicines makes it challenging to quickly and accurately identify the underlying cause and address the issue, further endangering patient safety.

### 6.3. Relatively Slow Treatment Effect

Compared to chemically synthesized drugs or surgical treatments, the therapeutic effects of natural medicines are generally milder and require a longer treatment course to achieve noticeable therapeutic outcomes. This characteristic stems from the “gentle regulation” mechanism of action of natural medicines and also determines the limitations in the onset of efficacy. Therefore, when used in treating diabetic nephropathy, natural medicines are more suitable for patients with relatively stable disease conditions. However, during acute disease flares or in severe cases, their therapeutic effects are far inferior to those of chemically synthesized drugs and cannot replace the emergency rescue function of surgical interventions.

Overall, the complexity of quality control, safety uncertainty, and delayed onset of efficacy collectively constitute the core bottlenecks in the development of natural medicines. These bottlenecks not only hinder their standardized, large-scale development but also limit their application in critical clinical scenarios to some extent. However, they precisely point the direction for technological innovation: by establishing precise quality control systems, developing toxicity detection technologies, and optimizing dosage forms, we can gradually overcome these bottlenecks.

## 7. Technological Innovations and Solutions

Natural compounds exhibit unique potential in the treatment of DKD, but issues such as low bioavailability, limited targeting precision, and safety concerns have long hindered their clinical translation. In recent years, the establishment of a translational roadmap centered on “nanodelivery platforms–formulation redesign–synthetic biology” (Figure 5), coupled with an R&D model supported by “multi-omics analysis–network pharmacology–AI-driven target prediction,” offers multidisciplinary solutions to overcome these bottlenecks through cross-disciplinary innovation. Key practical approaches and challenges are outlined below.

This figure visually presents the key directions of current pharmaceutical technology innovations, which mainly focus on four core areas: nanodelivery systems, formulation technologies, biotransformation technologies, and biological research methods.

### 7.1. Development of a Transformation Roadmap: Multi-Technology Synergy in Overcoming Natural Compound Application Barriers

Addressing the core limitations of natural compounds, the stepwise application of nanodelivery technology, formulation redesign, and synthetic biology constitutes a translational roadmap spanning “delivery–formulation–structure.” This approach enhances the feasibility of clinical translation through specific technical optimizations.

#### 7.1.1. Nanodelivery Systems

Nanodelivery systems utilize carriers such as natural polymers, synthetic polymers, or exosomes to load natural compounds through mechanisms like hydrophobic interactions and electrostatic adsorption. By regulating particle size and surface charge, these systems achieve controlled drug release and targeted accumulation [183,184]. Through modification of carrier materials and structural design, they address issues such as poor water solubility and susceptibility to degradation in vivo, while simultaneously enhancing renal targeting efficiency [185]. For instance, curcumin and epigallocatechin gallate exhibit therapeutic potential for diabetic nephropathy, but both suffer from poor water solubility and low bioavailability. Free curcumin exhibits a solubility of merely 1.268 ± 0.120 μg/mL, whereas shellac and locust bean gum-coacervated curcumin/epigallocatechin gallate nanoparticles (CESL-NP) exhibit a solubility of 75.833 ± 1.896 μg/mL—nearly 60-fold higher. In STZ-induced diabetic nephropathy mouse models, CESL-NPs significantly reduced fasting blood glucose, creatinine, and blood urea nitrogen levels while improving renal and pancreatic function [186]; PLA nanoparticles loaded with *Tinospora cordifolia* Willd. (TC-PLA NPs) exerted renal protective effects in STZ-induced diabetic nephropathy rats by reducing the expression of inflammatory factors (TNF-α, IL-6) and stabilizing renal function indicators (Scr, BUN) [64].

Despite the remarkable efficacy of nanodelivery systems, technical limitations persist: While PLA-PEG nanomicelles can enhance targeting through surface modification, they lack DKD-specific ligands (such as molecules binding to kidney epithelial cell-specific receptors) [187,188,189,190]. Although exosome delivery systems exhibit excellent biocompatibility, they face challenges such as low extraction yields and targeting efficiency that is susceptible to interference from the in vivo microenvironment. Furthermore, empirical studies directly employing natural drugs for DKD treatment remain scarce [191,192,193,194].

#### 7.1.2. Formulation Redesign

Beyond addressing natural compound delivery challenges through carrier-based nanodelivery systems, formulation redesign further breaks through application bottlenecks by improving dosage forms: Solid dispersion technology reduces drug crystallinity to enhance natural compound solubility [195]; liposome technology constructs bilayer membranes for controlled release to prolong the drug’s in vivo circulation time [196]; lysosomal conjugates utilize biomolecular specificity to achieve kidney-targeted delivery [197]; and combination formulations of natural compounds reduce the dosage of individual components and mitigate toxicity risks [177].

Specifically, addressing the low bioavailability of berberine due to poor solubility and insufficient membrane permeability, a highly bioavailable berberine solid dispersion formulation increases its bioavailability by 4-fold through inhibiting drug crystallization. In the db/db mouse DKD model, it effectively improves glomerular mesangial proliferation and renal function impairment [91]. Another example involves glycyrrhetinic acid liposomes loaded with carthamin yellow, which facilitate controlled release of carthamin yellow while enhancing drug stability, reducing side effects, and improving pharmacokinetic properties. This formulation effectively alleviates renal interstitial fibrosis in STZ-induced DKD rats [198]. Furthermore, baicalin–lysozyme conjugates leverage the specific affinity of lysosomes for renal tissue to elevate local drug concentrations in the kidney, significantly enhancing therapeutic efficacy in STZ-induced DKD rats [199]. Notably, a combined formulation of ursodeoxycholic acid and empagliflozin reduced urinary tract infections caused by high-dose empagliflozin monotherapy while lowering blood glucose levels in DKD model rats, and demonstrated superior inhibition of renal fibrosis compared to either drug alone [170].

However, formulation redesign still faces challenges at the clinical translation level: research on using lysosomal conjugates for the targeted treatment of diabetic kidney disease remains scarce, and there is a lack of data on combined use with commonly used DKD therapies such as SGLT2 inhibitors and ACE inhibitors. Standardized methods for optimizing the composition ratios in compound formulations are absent, making it difficult to ensure batch-to-batch consistency in therapeutic efficacy.

#### 7.1.3. Bioconversion and Synthetic Biology

Beyond delivery optimization and formulation improvements, bioconversion and synthetic biology technologies provide innovative support for transformation roadmaps by leveraging the structural and functional properties of natural compounds: solid-state fermentation of edible fungi converts active ingredients in traditional Chinese medicine, enhancing their water solubility and absorption efficiency [200]; synthetic biology techniques modify the structures of natural compounds to retain activity while reducing toxicity [201].

For example, the efficacy of *Astragalus membranaceus* enhanced through solid-state fermentation with *Paecilomyces cicadae* demonstrated superior renal protective effects in STZ-induced DKD mouse models by regulating gut microbiota balance and protecting podocytes [93,202]; for natural compounds like celastrol, which possesses both activity and toxicity, demethylzeylasteral retains therapeutic effects on DKD rats through structural modification while reducing side effects [182]. Meanwhile, selenized tripterine phytosomes overcome application bottlenecks in high-glucose-induced podocyte models by enhancing water solubility and imparting sustained-release properties [83].

However, this technological approach also faces pressing challenges: the types and concentrations of microbial metabolites produced during solid-state fermentation are highly susceptible to environmental factors, and standardized fermentation processes remain elusive [203]. Furthermore, the long-term safety and in vivo metabolic pathways of synthetic biology-modified natural compound analogs remain unclear, necessitating additional toxicological data to support clinical translation.

### 7.2. Multi-Technology Integration Supports Precision Development of Natural Medicines for DKD

Building upon the resolution of core application barriers for natural compounds through transformation roadmaps, a multi-technology-integrated R&D model further supports the efficient development of natural medicines for DKD by addressing their complex composition and ambiguous mechanisms of action. By integrating multi-omics analysis, network pharmacology, and AI-based target prediction, a predictive model system linking “constituents-targets-pathways-diseases” has been established, providing technical assurance for drug screening and mechanism elucidation.

#### 7.2.1. Synergistic Application of Multi-Omics and Network Pharmacology

This collaborative approach employs multi-omics technologies—including transcriptomics, metabolomics, and lipidomics—to first identify differentially expressed molecules (genes, lipids, metabolites) in the DKD pathological state. It then integrates network pharmacology to construct association networks linking natural compounds, potential targets, and disease pathways, thereby screening core functional targets. Finally, efficacy is validated through in vitro cell and in vivo animal experiments.

For instance, Xiao et al. predicted 85 potential targets for total berberine alkaloids via network pharmacology. Integrating STZ-DKD rat transcriptome data (121 differentially expressed genes), they identified AGEs-RAGE-TGFβ/Smad2 and PI3K-Akt as core pathways. Subsequent experiments further validated that modulating these pathways alleviates renal injury and fibrosis [204]. Similarly, Zhang et al. employed a combined approach of transcriptomics, network pharmacology, machine learning, and molecular docking/simulation techniques to reveal that *Berberis integerrima* targets 10 key genes through six active components. By synergistically regulating multiple signaling pathways, it exerts a protective effect against DKD [205].

However, this collaborative application model still faces practical bottlenecks: the fragmented nature of multi-omics data sources (such as inconsistent gene annotation standards across databases) complicates data integration and cross-validation [206]. Additionally, network pharmacology predictions yield false positives in target identification and pathway analysis, necessitating extensive experimental validation to narrow down candidates—significantly increasing R&D costs [207].

#### 7.2.2. AI Target Prediction: A Critical Complement to Precision Screening

As a crucial complement to multi-omics and network pharmacology, AI-based target prediction technology enhances the precision of natural product screening through its efficient data processing capabilities. By integrating the structural data of natural compounds (e.g., from the PubChem database), DKD-related target data (e.g., from the OMIM database), and multi-omics differential molecular data, target prediction models are constructed. This approach enables the rapid matching of natural compounds with core DKD targets and the prediction of their binding affinities. [123,208].

Currently, Hakime Öztürk et al. have developed an end-to-end deep learning model named DeepDTA based on convolutional neural networks, which can predict drug-target affinities and provides an efficient tool for drug screening [209]. In the future, integrating DKD pathological microenvironment data (such as inflammatory factor concentrations and pH levels) to optimize the model could enable dynamic matching predictions across “compound–target–microenvironment.”

However, the application of current AI target prediction models remains constrained by data quality: model training relies on high-quality annotated data, yet natural compound activity data suffers from inconsistent annotation and insufficient sample sizes.

In summary, the translational roadmap integrating nanodelivery systems, formulation redesign, and synthetic biology constructs holds promise for systematically overcoming clinical translation barriers of natural compounds. Meanwhile, the R&D model integrating multi-omics, network pharmacology, and AI-based target prediction provides the technological foundation for efficient development of natural medicines for DKD. The industry currently faces challenges including insufficient targeting efficiency of nanocarriers, unclear safety profiles of synthetic biology-modified products, and limited data quality in AI models. Future efforts should focus on establishing standardized technical processes, conducting large-scale multicenter clinical studies, and building interdisciplinary data-sharing platforms. These measures will advance natural drug development toward precision, controllability, and clinical applicability, thereby generating more novel therapeutic candidates for DKD treatment.

## 8. Clinical Practice of Natural Medicines

However, current clinical treatment regimens for DKD still have limitations such as insufficient therapeutic specificity and poor long-term safety. Natural products, with their unique advantages of multi-target regulation and good safety, are gradually becoming an important direction for complementary treatment of DKD. The following section will summarize and analyze natural products with potential applications based on specific clinical research data, providing references for clinical treatment selection and future research directions (Table 9).

**Table 9 ijms-26-11637-t009:** Current Clinical Trial Evidence Supporting the Use of Natural Products for Treating DKD.

Natural Products	Conditions	Dose	Test Duration	Primary Outcome	Reference
*Abelmoschus manihot*	2054 patients with CKD and proteinuria (≥150 mg/d)	12 years: 2.5 g TID; 6 to 12 years: 1.5 g TID; 2 to 6 years: 1 g TID	24 weeks	Proteinuria ↓	[210]
*Abelmoschus manihot*	413 patients with T2DM and DKD	2.5 g TID	24 weeks	Urine albumin-to-creatinine ratio ↓	[211]
*Triptery* *gium wilfordii hook f* extract	65 patients with T2DM and DKD who had proteinuria levels ≥ 2.5 g/24 h and serum creatinine levels < 3 mg/dL	120 mg daily for 3 months, followed by 60 mg daily for 3 months.	6 months	Proteinuria ↓	[212]
Resveratrol	60 patients with T2DM and DKD	500mg daily	90 days	Albuminuria ↓	[213]
Zicuiyin	88 patients with T2DM and DKD	crude drug amount 75 g, 150 mL, BID	8 weeks	eGFR ↑	[214]
Qidan Tangshen Granule	219 patients with T2DM and DKD	-	3 months and 12 months	Hemoglobin A1c and albumin-to-creatinine ratio ↓	[215]
Turmeric	40 patients with T2DM and DKD	22.1 mg, TID	2 months	Proteinuria ↓, TGF-β ↓, IL-8 ↓	[216]

Notes: DKD, diabetic kidney disease; CKD, Chronic Kidney Disease; T2DM, Type 2 Diabetes Mellitus; TID, three times daily; BID, twice daily; eGFR, estimated glomerular filtration rate; TGF-β, transforming growth factor-β; IL-8, interleukin 8; ↓, negative regulation, downregulate, inhibit; ↑, positive regulation, upregulate, promote.



*Abelmoschus manihot*



As the most well-established natural product for DKD treatment, it has been validated through multicenter clinical trials and long-term data and has been approved by the National Medical Products Administration for the treatment of chronic nephritis. It has a clear clinical application basis and is widely used in DKD treatment.

In patients with early-stage DKD, it demonstrated superior efficacy in reducing urinary albumin compared to losartan (50 mg/day), and the combination therapy with losartan showed significantly better efficacy than monotherapy [123]; A study involving 413 patients with type 2 diabetes (T2D) and DKD confirmed that its combination with irbesartan effectively reduces albuminuria and proteinuria [211]; further analysis of 2054 patients with CKD showed that the drug can reduce proteinuria while preserving kidney function, making it suitable for patients at different stages of kidney disease [210].

The existing evidence is based on multicenter, large-sample studies and includes long-term validation data, indicating a high level of evidence. However, data on the efficacy and safety of treatment for patients with end-stage DKD remain limited, and further studies are needed to address this gap.
*Tripterygium wilfordii Hook. f.* Extract

It is primarily used for controlling proteinuria in patients with DKD, particularly in advanced stages and those with overt proteinuria. It can serve as an adjunctive option when conventional treatments are ineffective. It has a distinct effect on reducing urinary protein levels: not only does it reduce the rate of kidney function decline in patients with advanced DKD, but it also demonstrates superior efficacy to valsartan. It also demonstrates greater efficacy in reducing overt proteinuria and controlling proteinuria in patients with normal estimated glomerular filtration rate (eGFR). However, its safety profile is a notable weakness, with a higher incidence of adverse reactions in clinical use, which may limit its widespread application. Close monitoring of adverse reactions is necessary during treatment [212,217].

Current studies have only confirmed short-term efficacy, and there is insufficient data to support long-term efficacy, leaving a significant gap in the evidence chain. The prospective clinical trial conducted by Xu, Zhao, and others [218] may provide additional evidence for assessing its long-term efficacy and safety, but no conclusive results have been reached yet.
Zicuiyin Decoction

It demonstrates unique advantages in patients with DKD and declining eGFR, particularly in those requiring improvement in renal function and gut microbiota balance, offering a new treatment direction for specific populations. Its efficacy is comprehensive and significant: it effectively improves renal function (reduces serum creatinine, increases eGFR), alleviates albuminuria and clinical symptoms, while regulating intestinal microbiota; studies have confirmed that its efficacy is even superior to that of Huangqi Capsules, performing better in patients with declining eGFR. No significant adverse reactions were observed in clinical observations, indicating good safety and high patient acceptance [214].

Current conclusions are based solely on data from small-scale trials and lack large-scale multicenter validation; the stability of long-term efficacy remains unclear, and the underlying mechanisms of action of the drug have not been fully elucidated, necessitating further research.
Resveratrol

It is primarily used as adjunctive therapy to angiotensin II receptor blockers (ARBs) for controlling urinary albumin in patients with T2DM and DKD. Preliminary studies suggest it may reduce urinary albumin excretion in patients with DKD, offering an alternative approach to patients with inadequate response to ARB therapy. No specific adverse reactions have been reported in existing small-scale studies; however, safety data remain insufficient and require validation through larger-scale studies [213].

The current level of evidence in this field remains low, with existing studies featuring limited sample sizes. There is a lack of comparative efficacy data across different dosage gradients, and no long-term follow-up results to support its claims. The precise efficacy and mechanism of action require further elucidation.

Notably, although numerous preclinical studies confirm that resveratrol significantly improves various pathological indicators of diabetic kidney disease (such as creatinine, blood urea nitrogen, and urine albumin-to-creatinine ratio) [219,220], the results of existing human clinical trials on resveratrol intervention for DKD exhibit heterogeneity. This may be related to its low water solubility and poor oral bioavailability [221,222].
Qidan Tangshen Granule and Curcumin

Both are being explored as adjunctive therapies for DKD: Qidan Tangshen Granule targets patients with T2DM and DKD, while short-term curcumin supplementation targets patients with overt DKD in T2DM. Qidan Tangshen Granule may provide clinical benefits by reducing oxidative stress, improving blood glucose levels, and enhancing renal function, demonstrating potential as an adjunctive therapy; short-term curcumin supplementation can reduce proteinuria and inflammatory factors (TGF-β, IL-8) in patients with overt T2DM-related DKD. Safety data for Qidan Tangshen Granule remain incomplete, while no adverse reactions were observed in studies of short-term curcumin supplementation, indicating good safety. Evidence for Qidan Tangshen Granule remains at the preliminary efficacy observation stage, lacking in-depth mechanistic studies and long-term validation; curcumin lacks large-scale, long-term follow-up validation data, with efficacy stability and applicable population boundaries still unclear [215,216].
Other natural products

Clinical trials of natural products such as Liuwei Dihuang Pills [223] and Tangshen Formula [224,225] for DKD are gradually being conducted, but research progress is currently limited, and no mature evidence chain has been established. Their efficacy and safety still need to be further supported by subsequent research data.

The incidence of DKD continues to rise, and existing treatment options struggle to fully meet clinical demands, making the clinical translation of natural products increasingly urgent. However, most studies in this field remain at the preliminary exploration or small-sample validation stage, facing significant limitations: on one hand, there is a lack of unified standards for precise efficacy assessment—differences in endpoint indicators (such as urine albumin-to-creatinine ratio, estimated glomerular filtration rate) and evaluation timepoints across studies make it difficult to compare results horizontally and integrate them systematically; on the other hand, the elucidation of mechanisms of action remains at the level of “phenomenological description.” The key molecular targets and signaling pathway interactions through which natural products modulate DKD pathological pathways (such as AGEs-RAGE and NLRP3 inflammasome) have not been fully elucidated, failing to provide a reasonable explanation for therapeutic efficacy variations. Second, there is a severe lack of safety evidence. Existing studies predominantly focus on short-term efficacy observations, with scarce long-term safety follow-up data spanning one year or more. Coverage of critical safety information—such as effects on hepatic and renal metabolism and drug interactions—remains extremely limited. Industrial-level challenges are equally pronounced: the uncertainty in efficacy stemming from the aforementioned research limitations, coupled with the complex composition of natural products and difficulty in quality control, has resulted in low participation willingness among pharmaceutical companies capable of conducting multi-center clinical trials. This further exacerbates a vicious cycle where insufficient research evidence leads to limited industrial investment, making it even harder to accumulate relevant evidence.

Moreover, the inherent properties of natural substances pose inherent challenges for research: they are typically multicomponent mixtures containing various bioactive compounds that may exhibit synergistic or antagonistic effects. This not only complicates the identification of the basis for pharmacological efficacy but may also mask the specific effects of the core target compound, posing significant challenges for elucidating mechanisms of action and ensuring quality control.

To overcome existing bottlenecks, generate high-quality research outcomes, and provide patients with more effective treatment options, a multidimensional approach is required: on one hand, comprehensively elevate the level of evidence by expanding sample sizes, extending follow-up periods, clarifying dose–response relationships, optimizing formulation processes to enhance bioavailability, establishing standardized dose conversion systems, and strengthening multidimensional safety monitoring. On the other hand, leveraging new technologies to specifically address pain points such as insufficient sample size, unclear mechanisms, and inadequate monitoring, while emphasizing international collaboration and multidisciplinary integration; cross-regional, multicenter trials can reduce the limitations of single studies, and synergistic collaboration among basic medicine, clinical medicine, data science, and other disciplines can form a complete closed-loop from mechanism research through clinical trial design to outcome analysis, further enhancing research quality. Advancing research along these directions will provide more reliable support for the clinical application of natural products in DKD treatment.

## 9. Conclusions

Natural products have demonstrated distinct advantages in DKD intervention. Integrating core research evidence, their mechanisms of action can be summarized into three key patterns: First, they precisely target critical nodes in pathological pathways such as AGEs-RAGE and NLRP3 inflammasome, directly acting on core pathological processes. Second, they form multidimensional protective chains through cross-mechanism synergistic regulation. Third, they cover cross-linkages across multidimensional pathological stages, ultimately constructing a comprehensive renal protection network. Preclinical experiments not only validate that these natural products effectively alleviate typical symptoms like proteinuria and delay renal fibrosis progression but also, for the first time, clarify their specific mechanisms of action in iron death regulation and metabolism, podocyte protection, anti-inflammation, and anti-fibrosis. This provides crucial theoretical and experimental foundations for subsequent research.

However, current research still faces several core challenges: First, the overall level of evidence remains low, with limited clinical data and generally small sample sizes in existing studies, resulting in limited persuasiveness. Second, the commonly used STZ model exhibits significant differences from the chronic progression of human DKD. Combined with high-dose experimental designs in some studies, this further exacerbates the difficulties in clinical translation. Third, natural products inherently suffer from low bioavailability and insufficient targeting specificity. Their complex composition significantly complicates quality control and dose management. Finally, their multi-targeted mechanisms remain incompletely elucidated, with critical regulatory parameters—such as autophagy activation thresholds—not fully defined. This undoubtedly limits the depth of mechanistic studies and reduces the willingness of pharmaceutical companies capable of conducting multi-center clinical trials to participate.

Fortunately, a series of innovative achievements has emerged, providing crucial support for overcoming these bottlenecks. Examples include the “anti-ferroptosis–antioxidation–immunity” cascade regulatory framework, the “nanodelivery–formulation reconstruction–synthetic biology” synergistic translation roadmap, and the “multi-omics–network pharmacology–AI target prediction” precision R&D paradigm. These achievements lay the groundwork for research breakthroughs across multiple dimensions, including technology, modeling, and pathways.

Moving forward, natural product research in DKD intervention should prioritize the following directions: First, optimize clinical translation technologies by developing DKD-specific targeted nanocarriers while standardizing formulation production processes and synthetic biology-related techniques to enhance translational feasibility. Second, establish a precision R&D system by building interdisciplinary data-sharing platforms to integrate multi-domain research data, further refining AI-based target prediction and mechanism analysis models to boost R&D efficiency. Third, conduct phased, multi-center clinical trials to systematically validate candidate drug efficacy and safety while actively exploring the potential of combination therapies, providing more comprehensive evidence for clinical application. Fourth, deepen research on mechanisms of action to clarify key parameter thresholds (e.g., autophagy activity thresholds) and intermolecular interaction mechanisms. Ultimately, establish a quantitative linkage system connecting “target–biomarker–efficacy” to advance the transformation of natural products into precision therapeutics for DKD.

## Figures and Tables

**Figure 1 ijms-26-11637-f001:**
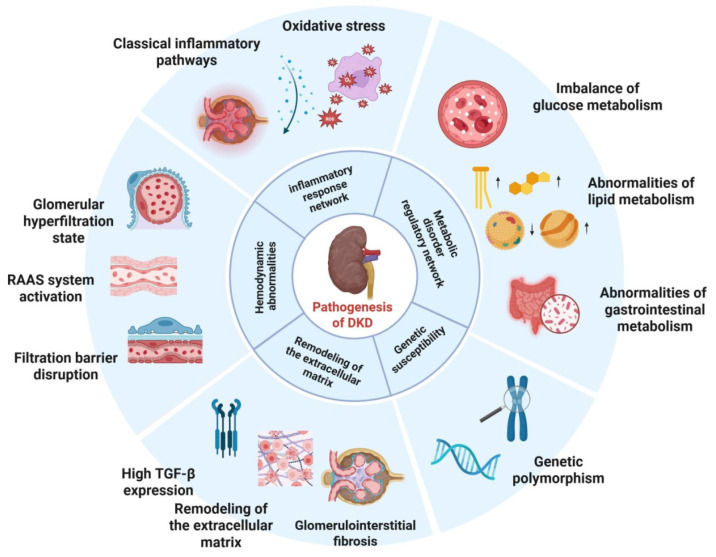
Schematic diagram of the pathogenesis of diabetic kidney disease (DKD).

**Figure 2 ijms-26-11637-f002:**
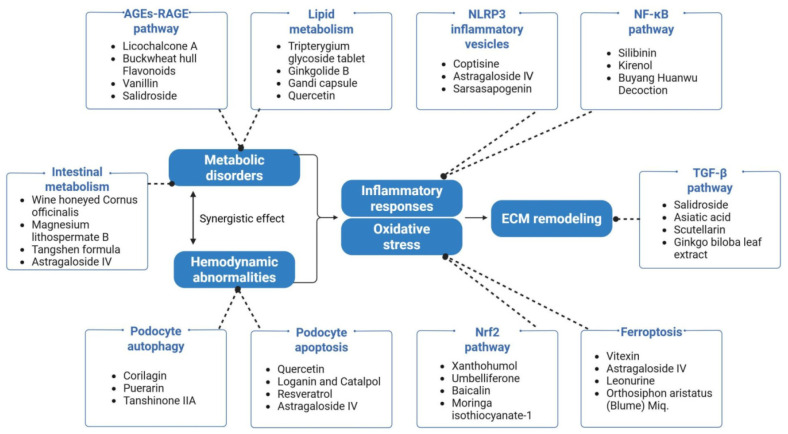
Schematic diagram of the key pathological cascade of diabetic kidney disease (DKD) kidney injury and the regulatory role of natural medicines.

**Figure 3 ijms-26-11637-f003:**
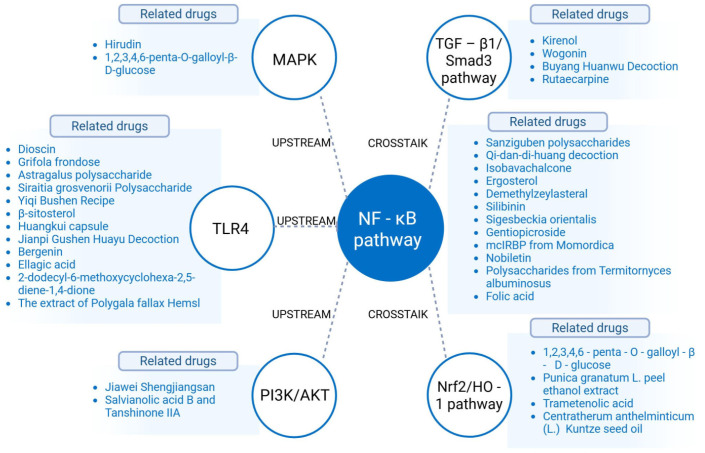
Schematic diagram illustrating the mechanism of natural medicines blocking inflammation-mediated kidney damage progression in diabetic kidney disease (DKD) via regulating the NF-κB pathway.

**Figure 4 ijms-26-11637-f004:**
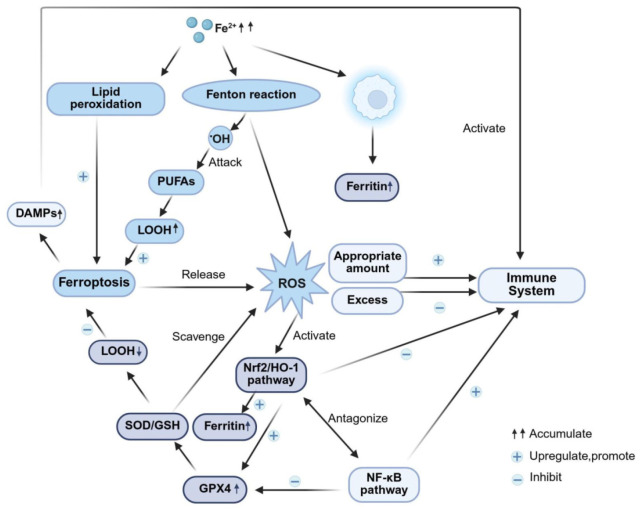
Schematic diagram illustrating the regulatory “antioxidation-ferroptosis-immunity” dynamic network of natural products.

**Figure 5 ijms-26-11637-f005:**
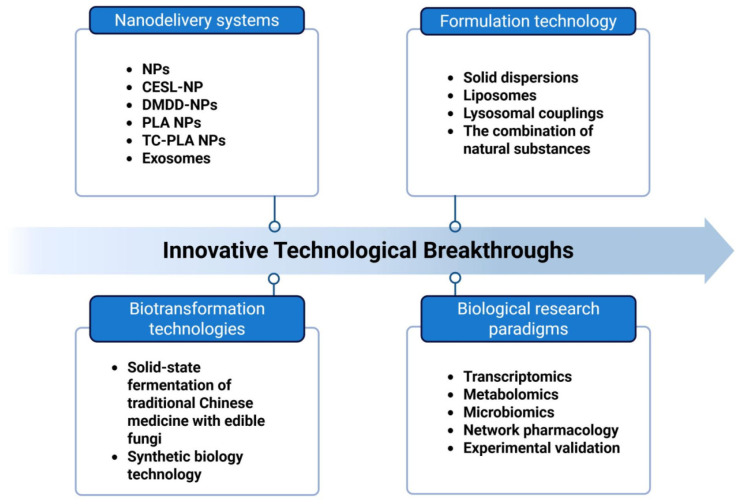
Schematic diagram illustrating the main areas of pharmaceutical technology innovations.

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
