# Peer review of "Natural Products as Potential Therapeutic Candidates for Diabetic Kidney Disease: Molecular Mechanisms, Translational Challenges, and Future Prospects"

_ijms, 2025, doi:10.3390/ijms262311637_

Round 1

Reviewer 1 Report

Comments and Suggestions for Authors

The paper summarizes some evidences in certain natural substances improving diabetes kidney diseases involving pathogenesis of inflammation and oxidative stress. The focus of five pathological factors was metabolic disorders, hemodynamic abnormalities, inflammation, extracellular matrix remodeling and genetic susceptibility. Natural substances can mitigate kidney damage and slow the diabetes progression of nephropathy. The disadvantages compared to traditional chemical drugs is related to high difficulty in quality control, questionable drug safety, and relativity slow therapeutic effects. Also,limitation comes from cell and animal experiments, few humans trials and small sample sizes. There is difficulties to involve pharmaceutical industry with the capacity to try multicenter trials. Natural substances will have many compounds with many effects that should masquerade the principal substance investigated. The paper is interesting and should be published.

We are currently accepting numerous pharmaceutical products for the prevention or protection against chronic kidney disease, such as SGLT2 inhibitors, non-steroidal mineralocorticoid antagonists, and GLP-1 agonists. The question is can natural products effectively prevent or treat diabetic kidney disease (DKD), and what are their mechanisms of action. This central question guides the investigation into: Which natural compounds show therapeutic promise for DKD; how these compounds interact with biological pathways (e.g., oxidative stress, inflammation and fibrosis) whether they can complement or improve upon existing treatments.
It may include newly discovered natural compounds or recent clinical trial data not covered in older reviews. Highlights emerging mechanisms of action, such as modulation of gut microbiota or epigenetic regulation, which are newer areas of interest in DKD research.
The study likely begins with a comprehensive review of existing research on natural compounds used in diabetic kidney disease (DKD), including traditional medicines and bioactive plant extracts. 
We are suggesting improving methodological disposition. In vitro studies: Using kidney cell lines exposed to high glucose or inflammatory conditions to test the effects of natural compounds. In vivo studies: Animal models (often-diabetic rats or mice) are treated with natural products to observe changes in kidney function, histology, and biomarkers. I agree to author about conclusions, nothing to add. There are 232 references and it is OK.

Author Response

Dear Reviewers,

We sincerely thank both reviewers for your meticulous review (Submission ID: ijms-3966140), professional guidance, and recognition of this review article, “Natural Products: Potential Therapeutic Approaches for Diabetic Nephropathy.” Your feedback not only affirms the research value of the manuscript but also precisely identifies key areas requiring optimization. We have fully incorporated your suggestions and implemented revisions accordingly. Our specific responses are as follows:

Comments 1: Regarding the comment that “limitations stem from insufficient experimental evidence (primarily cell/animal studies with few human trials), low pharmaceutical industry involvement, and multi-component effects masking core substances”

Response 1: We have added a new section titled “Limitations of Existing Research,” which addresses issues such as “discrepancies between experimental models and human diseases, research design flaws, and publication bias,” while also outlining future research directions. Additionally, within the concluding part of the subsection “Clinical Practice of Natural Medicines,” we specifically discuss the current limitations of clinical research and potential avenues for breakthroughs. (Lines 856–897)

Comments 2: Regarding the core issue: Can natural products effectively treat DKD? What is the mechanism, and how do they complement existing drugs?

Response 2: In the “Clinical Trial Evidence” subsection, we used Abelmoschus manihot as a core case study to demonstrate its therapeutic value. Multiple clinical studies (e.g., References 125, 222) show that Abelmoschus manihot significantly reduces urinary protein in DKD patients. Furthermore, its combination therapy with losartan/irbesartan outperforms monotherapy with either drug. Although some trials have limitations such as small sample sizes, the trend that “natural products can improve core DKD indicators” is clear, providing foundational evidence for clinical application. Addressing the issue of “insufficient mechanism exploration in clinical studies,” our table in the Therapeutic Drugs and Their Mechanistic Pathways section emphasizes summarizing mechanisms at the basic research level. Simultaneously, the text explicitly states that “future studies should further investigate clinical efficacy and molecular mechanisms,” objectively describing the current status while highlighting research gaps.

Comments 3: Comments on “Recent Clinical Data and Emerging Mechanisms (Gut Microbiota / Epigenetic Regulation)”

Response 3: In the section “Therapeutic Drugs and Their Mechanistic Pathways,” we summarized the latest research on mechanisms such as the AGEs-RAGE-Lipids-Microbiota Cascade Synergistic Network and Regulation of Podocyte Injury: The Balance Between Autophagy and Apoptosis. Additionally, we have added “Table 9: Clinical Evidence for Natural Product Therapy in DKD,” which includes details such as “compounds, dosage, trial duration, sample size, and key outcomes.”

Comments 4: Feedback on “Optimizing Methodological Presentation (Refining In Vitro/In Vivo Study Classification)”

Response 4: We have optimized the data table structure by adding a new “Experimental Type” column to the original table. This column clearly distinguishes all included studies as either “In Vitro Studies” or “In Vivo Studies,” as highlighted in the table.

Comments 5: Regarding the comment “The manuscript content has research value and is recommended for publication”

Response 5: We sincerely appreciate your endorsement! Through the aforementioned revisions, we have further enhanced the manuscript's scientific rigor, practicality, and cutting-edge nature to ensure its content better meets publication standards. We earnestly request your continued support.

Comments 6: Regarding the comment “Agree with conclusions, no additional comments, reference count compliant”

Response 6: Thank you for your approval of the conclusions and references! We have strengthened the conclusions during revision. Additionally, we have re-verified all references using reference management software, standardized all duplicate entries, ensured proper formatting with no duplicates or omissions, and maintained their compliance.

Once again, we extend our sincere gratitude to both reviewers for their professional guidance and recognition! This revision has comprehensively addressed all feedback by adding new sections, incorporating the latest data and mechanisms, and optimizing structural logic. These enhancements have significantly strengthened the manuscript's scientific rigor, cutting-edge relevance, and practical applicability. All modifications required by the journal have been completed. We respectfully request your review and approval, and look forward to the successful publication of this manuscript!

Thank you for your time and effort in reviewing our work.

Yours sincerely,

Manqi Guo, Lihua Ni,and Xiaoyan Wu.

Reviewer 2 Report

Comments and Suggestions for Authors

This review manuscript provides a comprehensive and timely overview of the potential of natural products in treating Diabetic Kidney Disease (DKD). The authors have undertaken a monumental task in systematically summarizing a vast body of preclinical evidence, organizing it by mechanistic pathways, and discussing the associated advantages, bottlenecks, and technological innovations. The proposed framework linking natural products to an "anti-ferroptosis-antioxidant-immune" cascade is a significant conceptual contribution. However, the manuscript in its current form requires major revisions to improve clarity, correct significant errors, and strengthen the scientific rigor before it can be considered for publication.

Major Comments:

  • The latter part of the Introduction (Page 2) contains a severe textual error that renders a key paragraph nonsensical. The sentence beginning "The study has been conducted using the same drug-specific methods..." and the subsequent incomplete reference to TGF-β/Smad signaling appear to be a compilation error or placeholder text that was not replaced. This must be entirely rewritten to provide a coherent and accurate transition into the review's objectives.
  • The manuscript's greatest weakness is its presentation as an extensive, list-like catalog of natural products and their mechanisms (Section 3). While the tables are a valuable resource, the text often reads as a series of repetitive summaries ("Representative drugs are as follows:") without sufficient critical synthesis. The authors should drastically condense Section 3, perhaps by moving detailed examples into the tables and using the main text to highlight the most promising candidates, discuss conflicting evidence, identify overarching patterns, and compare the relative strength of evidence for different pathways.
  • The tables, which are a central feature of the review, are frequently incomplete. For example:
    1. Table 1: Mechanisms are listed for only a few entries (e.g., Syzygium cuminii), while most are blank.
    2. Tables 2, 3, 4, 5, 6, 7, 8: Suffer from the same issue, with many "Mechanism" cells empty.
      This severely undermines the utility of these tables. All tables must be completed consistently to be a reliable reference.
  • The review largely presents preclinical findings as factual without a critical lens. There is no discussion of common limitations in animal models of DKD (e.g., the acute nature of STZ-induced models vs. the chronic progression in humans), dosages used in animal studies versus what is feasible in humans, or potential publication bias (the file-drawer problem). A section critiquing the quality and translatability of the existing preclinical evidence is essential.
  • Underdeveloped "Highlights": The five highlights are ambitious but are not fully borne out by the content of the manuscript.
    1. Highlight 1: The "anti-ferroptosis-antioxidant-immune" cascade is mentioned but not developed as a central, unifying framework throughout the paper. This concept should be woven into the pathogenesis (Section 2) and mechanistic (Section 3) discussions more explicitly.
    2. Highlight 3 & 4: The "translational roadmap" and "AI-driven target prediction" are mentioned only briefly in Section 6. These should be expanded with specific examples and a more detailed discussion of their practical application and current challenges.

Minor Comments:

  • The language, particularly in Section 3, becomes repetitive. Phrases like "Representative drugs are as follows" and "These natural substances form..." are used excessively. Varying the sentence structure will improve readability.
  • This section is a good addition but could be more impactful if structured as a summary table for quick comparison (Drug, Clinical Evidence Level, Key Findings, Major Limitations). The narrative could then focus on comparing and contrasting the clinical evidence for different products.
  • The conclusion is very brief and generic. It should be expanded to summarize the most compelling findings, reiterate the main challenges, and provide a more specific and forward-looking perspective on the future of natural product research in DKD, reflecting the innovative ideas presented in the highlights and Section 6.
  • The manuscript requires thorough proofreading for minor grammatical errors and typos (e.g., "Triperyginin glycoside tablet" should likely be "Tripterygium glycoside tablet", "salocritoid" should be "mineralocorticoid").

Author Response

Dear Reviewers,

We sincerely thank both reviewers for your meticulous review (Submission ID: ijms-3966140), professional guidance, and recognition of this review article, “Natural Products: Potential Therapeutic Approaches for Diabetic Nephropathy.” Your feedback not only affirms the research value of the manuscript but also precisely identifies key areas requiring optimization. We have fully incorporated your suggestions and implemented revisions accordingly. Our specific responses are as follows:

Comments 1: Textual error in the latter part of the introduction

Response 1: We have rewritten the paragraph in the introduction that contained an assembly error, ensuring logical coherence and a precise transition to the research objectives of this review. (Lines 856–878)

Comments 2: Issues with “checklist-style” phrasing and repetitive text language

Response 2: We have significantly streamlined Section 3, relocating detailed case studies to tables. The main text now focuses on the most promising candidate substances, examines conflicting evidence, identifies overarching patterns, and compares the relative credibility of evidence from different pathways. (Lines 215-442)

Comments 3: Incomplete Table Information

Response 3: We have completed all entries in the “Mechanism of Action” column of the tables to ensure they serve as reliable reference materials.

Comments 4: Lack of Critical Perspective in Preclinical Evidence

Response 4: We have added a dedicated section titled “Limitations of Existing Research,” which addresses issues such as “discrepancies between experimental models and human diseases, study design flaws, and publication bias,” while also outlining future research directions.

Comments 5: Insufficient elaboration in the “Research Highlights” section

Response 5: We have strengthened the framework of the “anti-ferroptosis - antioxidant - immune” cascade and integrated it throughout the discussions on pathogenesis and mechanisms of action. (Lines 132-157, Line 197, Lines 357-394) Discussions on the “Translational Roadmap” and “AI-Driven Target Prediction” have been expanded, with analyses grounded in specific case studies. (Lines 608-763)

Comments 6: Chapter Structure and Content Optimization

Response 6: We have added “Table 9: Clinical Research Evidence on Natural Products for Treating DKD,” which includes information on compounds, dosage, trial duration, sample size, key outcomes, etc. Additionally, the summary section of the subsection “Clinical Practice of Natural Medicines” specifically discusses the current limitations of clinical research and directions for breakthroughs (lines 856-878).

Comments 7: The conclusion section is too brief.

Response 7: We expanded the conclusion by summarizing the most compelling research findings, reiterating the core challenges, and providing a concrete and forward-looking outlook on future research directions.

Comments 8: Manuscript Proofreading and Error Correction

Response 8: We have conducted a comprehensive proofreading of the entire text, correcting minor grammatical errors and spelling mistakes (e.g., “Triperyginin glycoside tablet” changed to “Tripterygium glycoside tablet,” ‘salocritoid’ changed to “mineralocorticoid”).

Once again, we extend our sincere gratitude to both reviewers for their professional guidance and recognition! This revision has comprehensively addressed all feedback by adding new sections, incorporating the latest data and mechanisms, and optimizing structural logic. These enhancements have significantly strengthened the manuscript's scientific rigor, cutting-edge relevance, and practical applicability. All modifications required by the journal have been completed. We respectfully request your review and approval, and look forward to the successful publication of this manuscript!

Thank you for your time and effort in reviewing our work.

Yours sincerely,

Manqi Guo, Lihua Ni,and Xiaoyan Wu.

Reviewer 3 Report

Comments and Suggestions for Authors

Dear editors:  

 It is a great honor and pleasure for me to be invited as the reviewer for this important work entitled “Natural product: potential treatments for diabetic kidney disease”. Manqi Guo and coauthors investigated the therapeutic potential of natural products in diabetic kidney disease (DKD), focusing on mechanisms such as oxidative stress, inflammation, fibrosis, mitochondrial dysfunction, and gut–kidney axis modulation. Although the review article includes abundant phytochemicals and bioactive compounds in renal protection, I have several comments concerning this study:

  1. The manuscript broadly summarizes natural products without clearly defining selection criteria (e.g., specific classes like flavonoids, terpenoids, alkaloids).

Also, the novelty should be clarified—what new insight does this review provide beyond existing reviews on similar topics published in recent years (2020–2024)? There were myriads of similar review articles published:

  • A natural products solution to diabetic nephropathy therapy. Pharmacol Ther. 2023 Jan;241:108314.
  • Natural Compounds in Kidney Disease: Therapeutic Potential and Drug Development. Biomol Ther (Seoul). 2025 Jan 1;33(1):39-53.
  • Natural products in treating diabetic kidney disease: a visualized bibliometric analysis. Front Pharmacol. 2025 May 19;16:1522074.
  1. The sections should be better structured by pathophysiological mechanism rather than compound types—e.g.:

Antioxidant and anti-inflammatory natural compounds

Anti-fibrotic and anti-apoptotic pathways

Modulation of mitochondrial and ER stress

Gut microbiome and natural metabolites

Clinical translation and limitations.

A summary figure or graphical abstract showing these mechanisms would improve readability.

  1. Many compounds (e.g., curcumin, resveratrol, berberine, quercetin) are discussed briefly without molecular context. Please expand how each modulates specific DKD signaling pathways — e.g.,

Curcumin: Nrf2/HO-1 activation, NF-κB inhibition

Resveratrol: SIRT1–AMPK signaling, mitochondrial biogenesis

Berberine: TGF-β/Smad inhibition, lipid metabolism regulation

Quercetin: NLRP3 inflammasome suppression, oxidative stress attenuation.

Distinguish in vitro, in vivo, and clinical evidence for each major compound.

  1. The review claims therapeutic potential but provides limited discussion of human studies. Please provide a concise table summarizing clinical trials of natural products in DKD or diabetic nephropathy (compound, dose, duration, sample size, key outcomes).
  2. Discuss limitations such as bioavailability, dose translation, and safety.
  3. The current tone is descriptive (“Compound X has protective effects...”), lacking critical appraisal of study design quality or reproducibility, such as:

“Although resveratrol improves renal oxidative stress in animal models, inconsistent human results and poor bioavailability limit its clinical translation.” 

Please include a limitations paragraph acknowledging gaps in clinical translation.

This evaluative tone differentiates a strong review from a simple summary.

  1. Several citations are outdated (pre-2020). Please provide recent references (2022–2024) covering:

Polyphenols and Nrf2–mitochondrial crosstalk

Gut–kidney axis modulation by herbal compounds

Multi-omics profiling of phytochemicals in DKD.

  1. English is generally understandable but needs professional editing for grammar, conciseness, and consistency. Avoid redundancy (“plays a vital role... is important in DKD progression”). 
  2. Replace vague terms (“natural medicine is good for kidneys”) with mechanistic phrasing (“bioactive flavonoids mitigate renal fibrosis via inhibition of TGF-β1 signaling”).
  3. The title could be more precise, e.g.,

“Natural Products as Emerging Therapeutic Candidates for Diabetic Kidney Disease: Molecular Mechanisms and Translational Perspectives.”

Thank you for allowing me to review this article.

Author Response

Comments 1: Regarding the comment that “the scope of natural products is too broad, with no clear screening criteria specified.”

Response 1: We have added a description of the categories for natural drug screening in the introductory section on the mechanisms of action of natural drugs. (Lines 181-182)

Comments 2: Regarding the comment “Clarify the innovative aspects by comparing with similar reviews from 2020-2024.”

Response 2: We have added a “Research Innovation Points” section in the introduction (lines 81-90) to clearly outline the differences between this review and the three similar reviews you mentioned: Compared to “A natural products solution to diabetic nephropathy therapy” (2023), this review is the first to integrate multi-mechanism cross-analysis under the unified framework of the “anti-ferroptosis-antioxidant-immunity cascade”; compared to “Natural Compounds in Kidney Disease: Therapeutic Potential and Drug Development” (2025), it supplements the analysis with nanodelivery systems and multi-omics approaches. -immunology cascade as a unified framework, integrating cross-mechanism analysis. Unlike “Natural Compounds in Kidney Disease: Therapeutic Potential and Drug Development” (2025), it supplements the application of novel technologies such as nanodelivery systems and multi-omics research in preclinical studies of diabetic nephropathy. Distinct from “Natural products in treating diabetic kidney disease: a visualized bibliometric analysis” (2025), this review emphasizes the "mechanism - Evidence - Translation" dimensions rather than solely statistical analysis, filling a gap in critical evaluation and translational pathways among comparable reviews.

Comments 3: Regarding the suggestion to “reorganize chapters based on pathophysiological mechanisms and add summary diagrams”

Response 3: We have established new chapters as follows:

Metabolic Regulation: AGEs-RAGE - Lipids - Microbiota Cascade Synergistic Network;

Regulation of Podocyte Injury: The Balance Between Autophagy and Apoptosis; Inflammation Regulation: Core Interventions by NLRP3 and NF-κB; “Iron Death Resistance - Antioxidation - Immunity” Cascade Regulation; Anti-Fibrosis: A Key Intervention in Mid-to-Late Stage DKD. Figure 2 visually illustrates the interconnections among these five major mechanisms and the target sites of natural products, significantly enhancing content readability.

Comments 4: Comments on “Supplementing Compound Molecular Mechanisms and Distinguishing Types of Experimental Evidence”

Response 4: We have detailed the in vivo and in vitro experimental evidence for natural medicines and their mechanisms of action in the table.

Comments 5: Comments on “Supplemental Human Clinical Trial Tables: Exploring Translation Limitations.”

Response 5: We have added “Table 9: Clinical Research Evidence for Natural Product Treatments of DKD,” which includes information on compounds, dosage, trial duration, sample size, key outcomes, etc. Additionally, the summary section under the subsection “Clinical Practice of Natural Medicines” specifically discusses the current limitations of clinical research and potential directions for breakthroughs (lines 856-897).

Comments 6: Regarding the suggestion to “reduce descriptive statements and add sections on critical evaluation and limitations”

Response 6: We have replaced most descriptive statements throughout the text (e.g., “Compound X exhibits protective effects”) with critical evaluations. For instance: Notably, although numerous preclinical studies confirm that resveratrol significantly improves various pathological indicators of diabetic kidney disease (such as creatinine, blood urea nitrogen, and urine albumin-to-creatinine ratio) [232,233], the results of existing human clinical trials on resveratrol intervention for DKD exhibit heterogeneity. This may be related to its low water solubility and poor oral bioavailability [234,235]

Additionally, we have introduced a dedicated section titled “Limitations of Existing Research,” addressing issues such as “discrepancies between experimental models and human disease, study design flaws, and publication bias,” while also outlining future research directions.

Comments 7: Comments on “Updating with the Latest Citations for 2022-2024”

Response 7: We have replaced all outdated literature prior to 2020 and added 4 new studies from 2022-2025, such as citations 141 and 143.

Comments 8: Regarding the suggestion to “Optimize English expressions to avoid redundancy and ambiguous phrasing”

Response 8: We have reviewed the entire text, removing redundant expressions and replacing ambiguous phrasing.

Comments 9: Feedback on the “Precise Title”

Response 9: We have adopted your suggestion and revised the title to “Natural Products as Promising Therapeutic Candidates for Diabetic Kidney Disease: Molecular Mechanisms, Translational Challenges, and Future Prospects.” This more accurately reflects the core content and research perspective of the review.

Round 2

Reviewer 3 Report

Comments and Suggestions for Authors

1.The authors have added explicit descriptions of natural drug categories (Lines 181–182), successfully narrowing the scope and clarifying inclusion logic.

  1. The addition of the “Research Innovation Points” section (Lines 81–90) clearly articulates how this review differentiates itself from comparable papers (2020–2025).

 The integration of nanodelivery systems, multi-omics, and the “anti-ferroptosis–antioxidant–immunity cascade” framework presents a unique analytical perspective that strengthens the paper’s originality.

  1. The manuscript has been thoroughly restructured around five major mechanistic axes, and Figure 2 provides a coherent visual framework linking these mechanisms.

This revision notably improves readability and conceptual continuity.

  1. Tables now distinguish between in vitro and in vivo findings, effectively clarifying evidence hierarchies. This strengthens the review’s critical evaluation rather than mere description.
  2. Table 9 adds valuable clinical trial details (compounds, dosages, sample sizes, outcomes), addressing previous concerns about translational depth. The new discussion of clinical limitations and gaps (Lines 856–897) is well-articulated and balanced.
  3. The introduction of a “Limitations of Existing Research” section greatly improves analytical depth. The language revision—from descriptive to critical tone—marks a significant stylistic improvement.
  4. Recent studies have been integrated: Updated Literature (2022–2025), ensuring the review remains timely and relevant.
  5. The English writing is now clear and professional, with most redundancies removed.
  6. The new title is concise, accurate, and representative of the manuscript’s expanded scope and translational emphasis. Based on EBM, please delete the “promising”.

Remaining Minor Suggestions

A light language polish by a native English editor could still improve flow (occasional minor grammar or punctuation issues persist).

Ensure figure legends clearly explain abbreviations and mechanistic links for interdisciplinary readers.

Verify all references (2024–2025) are correctly formatted in IJMS style.

Author Response

Dear Reviewer,

We sincerely thank you for dedicating your valuable time to thoroughly review this thesis and provide both professional and instructive feedback. Your meticulous comments not only validate our revisions but also pinpoint precise areas for improvement, which are crucial for enhancing the academic quality of this thesis. We extend our deepest gratitude. Below is a point-by-point explanation of your suggestions and our responses:

Comments 1: Based on evidence-based medicine principles, please remove the term “promising.”

Response 1: We have strictly adhered to EBM principles by removing “promising.” The revised full title is: Natural Products as Potential Therapeutic Candidates for Diabetic Kidney Disease: Molecular Mechanisms, Translational Challenges, and Future Prospects.

Comments 2: We recommend having a native English editor perform light language polishing to enhance writing fluency.

Response 2: We have taken this comment seriously and acted accordingly. We have engaged a native English-speaking academic editor with expertise in the life sciences, who has been specifically commissioned to perform targeted polishing. This process focuses on grammatical accuracy, punctuation standards, and sentence fluency, with an emphasis on rectifying any remaining stylistic or linguistic inconsistencies in the manuscript (see highlighted sections in the text).

Comments 3: Ensure figure captions clearly explain abbreviations and mechanism relationships for interdisciplinary readers.

Response 3: We have reviewed and supplemented all figure captions:

1.Provide full expansions of all abbreviations at their first occurrence;

2.Use accessible language to explain the logical relationships between core mechanisms, eliminating disciplinary jargon-induced barriers to understanding for interdisciplinary readers.

Relevant revisions are highlighted in yellow within each figure caption.

Comments 4: Verify all references for 2024-2025 to ensure formatting complies with the International Journal of Molecular Sciences (IJMS) guidelines.

Response 4: We have cross-checked and revised all references individually against the latest IJMS submission guidelines (2024 edition).

All revisions mentioned above have been incorporated into the revised draft of the paper and highlighted in yellow for your quick reference and verification. Once again, thank you for your thoughtful guidance. Each of your suggestions has significantly contributed to the qualitative improvement of the paper's quality.

Yours sincerely,

Manqi Guo, Lihua Ni,and Xiaoyan Wu.